# Debiasing Diffusion Models via Score Guidance

**Piyush Tiwary**                                                     *piyushtiwary@iisc.ac.in*
*Department of Electrical Communication Engineering*
*Indian Institute of Science*

**Prabhav Verma**                                                    *prabhavverma@iisc.ac.in*
*Department of Electronic Systems Engineering*
*Indian Institute of Science*

**Prathosh A.P.**                                                    *prathosh@iisc.ac.in*
*Department of Electrical Communication Engineering*
*Indian Institute of Science*
*LatentForce*

**Reviewed on OpenReview:** *https://openreview.net/forum?id=vAz8xUHyTe*

## Abstract

With the increasing use of Diffusion Models (DMs) in everyday applications, it is very important to ensure that these models are *fair* towards various demographic/societal groups. However, due to several reasons DMs inherit biases towards specific gender, race and community, which can perpetuate and amplify societal inequities. Hence, it is important to *debias* DMs. Previous debiasing approaches require additional reference data, model fine-tuning, or auxiliary classifier training - each of which incur additional cost. In this work, we provide a training-free inference-time method for debiasing diffusion models. First, we provide a theoretical explanation for the cause of biases inhibited by DMs. Specifically, we show that the unconditional score predicted by the denoiser can be expressed as a convex combination of conditional scores corresponding to the attributes under consideration. We then argue that the weights allocated to underrepresented attributes are less which leads to domination of other attributes in overall score function. Building on this, we propose a score-guidance method that adheres to a user provided reference distribution for generation. Moreover, we show that this score guidance can be achieved via different modalities like 'text' and 'exemplar images'. To our knowledge, our method is the first to provide a debiasing framework that can utilize different modalities for diffusion models. We demonstrate the effectiveness of our method across various attributes on both unconditional and conditional text-based diffusion models, including Stable Diffusion.

## 1 Introduction

Recent advancements in diffusion models (DMs) (Ho et al., 2020; Nichol & Dhariwal, 2021; Sohl-Dickstein et al., 2015; Song et al., 2021) have garnered significant attention across various domains, including medical diagnosis (Rahman et al., 2023; Zhan et al., 2024; Chen et al., 2024; Zbinden et al., 2023; Tiwary et al., 2025a; 2024), drug discovery (Levy & Rector-Brooks, 2023; Alakhdar et al., 2024), material analysis (Lei et al., 2024; Yang et al., 2024), video generation (Xing et al., 2024; NVIDIA, 2025; Bar-Tal et al., 2024; Ho et al., 2022; Brooks et al., 2024; Zhang et al., 2023b), and AI assistants (Higham et al., 2023). Many of these applications are highly sensitive, where the presence of inherent biases can lead to serious societal implications. Furthermore, with the increasing adoption of AI technologies, regulatory frameworks such as GDPR (Voigt & Von dem Bussche, 2017) impose stringent requirements for fairness and transparency, making the mitigation of biases in these models even more critical.

Despite their potential, DMs have been shown to exhibit notable biases related to attributes such as gender and race (Perera & Patel, 2023; Luccioni et al., 2023; Rosenberg et al., 2023; Mandal et al., 2023; Schramowski et al., 2023; Zhang et al., 2023a; Tiwary et al., 2025b). These biases can arise from multiple sources. A primary contributor is dataset bias — where the training data fails to accurately represent real-world distributions. However, studies have demonstrated that even when models are trained on balanced datasets, biases can still emerge (Perera & Patel, 2023). Other factors contributing to these biases include labeling biases (Cabrera et al., 2014) and cultural biases (Peters & Carman, 2024). Addressing these challenges highlights the importance of developing effective methods for debiasing diffusion models.

Existing approaches for debiasing often rely on access to a reference dataset (Zhang et al., 2023a) or require fine-tuning the pre-trained DMs (Choi et al., 2020; Xu et al., 2018b; Yu et al., 2020). As an alternative, Parihar et al. (2024) proposed a novel and generic reference distribution-based debiasing framework. In this method, users specify a desired reference distribution $\mathbf{p_{ref}^a}$ for a particular attribute $\mathbf{a}$, and the generated samples from the DM are adjusted to reflect this target distribution. This is achieved by minimizing the divergence between $\mathbf{p_{ref}^a}$ and a surrogate distribution $\mathbf{p_\theta^a}$, where $\mathbf{p_\theta^a}$ is estimated from generated samples using an auxiliary classifier parameterized by $\theta$. While this approach avoids the need for a reference dataset or fine-tuning of DMs, it introduces additional complexity by requiring the training of an auxiliary classifier to estimate $\mathbf{p_\theta^a}$. Further, it also requires access to training set for training the auxiliary classifier which might not be possible.

In this work, we tackle the problem of debiasing diffusion models from first principles. By leveraging Tweedie's formula (Efron, 2011), we demonstrate that the (Stein's) score function provided by DMs can be expressed as a weighted average of conditional scores associated with concerned attributes. We hypothesize that underrepresented attributes receive lower weights in this formulation, leading to their underrepresentation in generated samples. This insight enables us to address bias in a training-free manner, without relying on auxiliary classifiers or fine-tuning. The problem reduces to 'nudging' or 'guiding' generated samples toward the provided reference ratio, counteracting the inherent weight ratio learned by DMs. We employ Tweedie's formula to guide the conditional scores of an appropriate number of samples to align with the provided reference distribution. Our framework accommodates multiple modalities, including text and exemplar images, for this score guidance. For text-based guidance, we utilize off-the-shelf large-scale pretrained models like CLIP (Radford et al., 2021a), while for exemplar images, we develop a framework to guide scores toward desired conditional modes.

The simplicity and generality of our framework position it to drive future research in fair generation. Our main contributions are:

1. A novel first-principles formulation revealing inherent biases in diffusion models, utilizing Tweedie's formula to demonstrate how the score predicted by DMs comprise weighted averages of conditional scores, with underrepresentation stemming from lower assigned weights.

2. We reduce the debiasing problem to score guidance, where we demonstrate that fair generation can be achieved by systematically adjusting the scores of a calculated proportion of samples to match the desired reference distribution, effectively rebalancing the representation of different attributes without modifying the underlying model

3. We provide a training-free framework that enables flexible debiasing through either text prompts or exemplar images, allowing users to guide pretrained diffusion models using natural language descriptions via CLIP embeddings or visual examples through our novel score-based guidance method, without requiring any additional training or model modifications.

4. We perform thorough experiments on state-of-the-art DMs and demonstrate effectiveness of proposed method in fair generation.

Table 1: Comparison of different guidance and debiasing methods. By stacking the citation below the method name, we improve readability.

| Methods | Train. Free | Infer. Time | Train Data | Uncond. + Cond. | Multi- Modal. | Score Guide. |
|---|---|---|---|---|---|---|
| *Generic Guidance Methods* | | | | | | |
| **Universal Guidance** [Bansal et al., 2023] | ✗ | ✓ | ✓ | ✓ | ✓ | ✓ |
| **Latent Editing** [Kwon et al., 2022] | ✓ | ✓ | ✗ | ✓ | ✗ | ✗ |
| *Specialized Debiasing Methods* | | | | | | |
| **Fair Diffusion** [Friedrich et al., 2023] | ✓ | ✓ | ✗ | ✗ | ✗ | ✗ |
| **Fair Mapping** [Li et al., 2024] | ✗ | ✓ | ✗ | ✗ | ✗ | ✗ |
| **ITI-Gen, ADFT** [Zhang et al., 2023a; Shen et al., 2024] | ✗ | ✗ | ✗ | ✗ | ✗ | ✗ |
| **UCE, TIME, MIST** [Gandikota et al., 2024; Orgad et al., 2023...] | ✗ | ✗ | ✗ | ✗ | ✗ | ✗ |
| **Balancing Act** [Parihar et al., 2024] | ✗ | ✓ | ✓ | ✓ | ✗ | ✗ |
| **Ours (SG)** | ✓ | ✓ | ✗ | ✓ | ✓ | ✓ |

## 2 Related Works

### 2.1 Biases in Generative Models

Generative models inherit and propagate biases from their training data, making fairness in synthetic data a critical concern for downstream applications. Gender biases, for instance, are evident in diffusion models trained on CelebA-HQ Parihar et al. (2024); Tiwary et al. (2025b) and persist in large-scale models like Stable Diffusion Friedrich et al. (2023); Zhang et al. (2023a). This issue is widespread, affecting GANs Yu et al. (2020); Humayun et al. (2021); Xu et al. (2018a), LLMs Gehman et al. (2020); Abid et al. (2021); Bender et al. (2021); Ding et al. (2022), and VLMs Zhang et al. (2022). As manual debiasing of massive training datasets is intractable, post-training fairness interventions are a crucial approach for deployed systems.

### 2.2 Debiasing Diffusion Models

Recent works have made significant strides in addressing bias in Diffusion Models (DMs) (Friedrich et al., 2023; Zhang et al., 2023a; Shen et al., 2024; Parihar et al., 2024; Li et al., 2024). Fair Diffusion (Friedrich et al., 2023) introduces an approach utilizing a look-up table to identify bias-prone concepts within prompts (such as gender associations with occupations like firefighter). Their method incorporates a correction term into the prompt embeddings to balance the representation between under- and over-represented classes. Similarly, ITI-Gen (Zhang et al., 2023a) furthers this concept by learning token embeddings for biased concepts through reference images, which are then appended to the original prompt embeddings. On similar lines, Fair Mapping (Li et al., 2024) introduces a lightweight fine-tuning by adding a linear layer in the CLIP model to equalize the odds of each attribute embedding in a given concept class. This is done by ensuring that class embedding is equi-distant from the attribute embeddings for each attribute. While these approaches have shown promise, their reliance on text embedding manipulation limits their applicability to conditional DMs only. Further, there are few editing methods like UCE (Gandikota et al., 2024), TIME (Orgad et al., 2023) and MIST (Yesiltepe et al., 2024) manipulate the cross-attention layers between feature maps and text embeddings to balance the representation of individual attributes. ADFT (Shen et al., 2024) proposed an adjusted direct finetuning of T2I models to align with a reference distribution by optimizing the text tokens. However, it again requires finetuning of model, moreover, it requires significant cost to compute adjusted gradients required for their method. Recent efforts have also explored debiasing directly within the

text conditioning space. For instance, LightFair (Han et al., 2025) provides an efficient debiasing method by directly mitigating biases within the pre-trained text encoders (e.g., CLIP) of Text-to-Image models. Consequently, its application is fundamentally restricted to conditional diffusion models.

A more comprehensive framework was proposed by Parihar et al. (2024), addressing both conditional and unconditional DMs. Their method introduces a reference distribution-based debiasing approach where users specify a desired distribution ($\mathbf{p_{ref}^a}$) over attribute classes, which then guides the sampling process to maintain these proportions. The framework leverages the **H**-space flexibility of the denoiser (Kwon et al., 2022), utilizing a lightweight classifier that processes batches of **H**-space vectors to produce a softmax distribution ($\mathbf{p_\theta^a}$) over attribute classes. This classifier is trained using DDIM Inversion (Song et al., 2021; Mokady et al., 2023) on the training data. During generation, the framework optimizes **H**-space vectors to minimize the KL-Divergence between $\mathbf{p_{ref}^a}$ and $\mathbf{p_\theta^a}$. However, this approach faces several limitations: it requires training an additional classifier, demands access to labeled training data used in the original DM training, and cannot effectively utilize alternative curated datasets due to potential DDIM Inversion errors that could compromise classifier training reliability.

Our work addresses these limitations by introducing a novel score-guidance method that eliminates additional training requirements and operates without access to training data. While we build upon the reference distribution-based debiasing framework of Parihar et al. (2024), we diverge from their distribution guidance approach. Instead, we propose a more versatile inference-time 'score-guidance' method that supports fair generation through multiple modalities, including text and exemplar images. This approach offers greater flexibility while maintaining effectiveness in bias mitigation.

### 2.3 Guidance in Diffusion Models

Guidance is a well-studied technique in DMs used for various objectives, including debiasing Parihar et al. (2024); Agarwal et al. (2023); Chefer et al. (2023); Um & Ye (2024). Methods like Universal Guidance Bansal et al. (2023) use external models (e.g., classifiers) to guide the entire diffusion process. However, this approach can suffer from erroneous guidance, as external models must be robust to noisy inputs, especially during early diffusion steps. In contrast, our work applies guidance selectively only within specific time windows, informed by recent findings on feature emergence Li & Chen (2024); Choi et al. (2022); Raya & Ambrogioni (2024); Georgiev et al. (2023). Table 1 provides a comparison of these methods.

## 3 Proposed Methodology

### 3.1 Preliminaries

Diffusion Models (Ho et al., 2020; Sohl-Dickstein et al., 2015; Song & Ermon, 2019; Song et al., 2021) are probabilistic generative models defined by two key processes. The forward process systematically corrupts data points into isotropic Gaussian noise through a predefined noise schedule. The reverse process learns to iteratively denoise a Gaussian random variable to generate samples from the training distribution. This denoising is achieved by training a model to estimate the noise that should be removed at each timestep. These processes are formally defined by their transition kernels:

$$q(\mathbf{x}_t|\mathbf{x}_{t-1}) = \mathcal{N}(\mathbf{x}_t; \sqrt{\alpha_t}\mathbf{x}_{t-1}, (1-\alpha_t)I) \tag{1}$$

$$p_\theta(\mathbf{x}_{t-1}|\mathbf{x}_t) = \mathcal{N}(\mathbf{x}_{t-1}; \mu_\theta(\mathbf{x}_t), (1-\alpha_t)I) \tag{2}$$

where $\alpha_t$ represents the predetermined noise schedule parameters. The denoising model is optimized using the following objective:

$$\theta^* = \arg\min_\theta \mathbb{E}_{t,\mathbf{x}_t,\epsilon} \left[ |\epsilon - \epsilon_\theta(\mathbf{x}_t, t)|_2^2 \right] \tag{3}$$

## 3.2 Bias in Diffusion Models

The objective in Eq. 3 can be interpreted through score-matching (Song et al., 2021; Luo, 2022; Chan et al., 2024). Using Tweedie's formula, we can express the score function at time $t$ in terms of $\epsilon_\theta(\mathbf{x}_t, t)$[1], yielding:

$$\hat{\mathbf{x}}_{0|t} \triangleq \mathbb{E}\left[\mathbf{X}_0 \mid \mathbf{X}_t = \mathbf{x}_t\right] = \frac{\mathbf{x}_t + (1 - \bar{\alpha}_t)\nabla \log p(\mathbf{x}_t)}{\sqrt{\bar{\alpha}_t}} \tag{4}$$

$$= \frac{\mathbf{x}_t - \sqrt{1 - \bar{\alpha}_t}\epsilon_\theta(\mathbf{x}_t, t)}{\sqrt{\bar{\alpha}_t}} \tag{5}$$

Following (Song et al., 2021; Dieleman, 2023), we can interpret $\hat{\mathbf{x}}_{0|t}$ as an estimate of the final denoised sample $\mathbf{x}_0$ given the current noisy sample $\mathbf{x}_t$. This leads to:

$$\mathbb{E}\left[\mathbf{X}_0 \mid \mathbf{X}_t = \mathbf{x}_t\right] = \mathbb{E}\left[\mathbb{E}\left[\mathbf{X}_0 \mid \mathbf{X}_t = \mathbf{x}_t, \mathbf{Y}\right]\right] \tag{6}$$

$$= \sum_{\mathbf{a}_i} p(\mathbf{a}_i)\mathbb{E}\left[\mathbf{X}_0 \mid \mathbf{X}_t = \mathbf{x}_t, \mathbf{Y} = \mathbf{a}_i\right] \tag{7}$$

Combining Eq.4 with Eq.7 using conditional and unconditional scores gives:

$$\nabla \log p(\mathbf{x}_t) = \sum_{\mathbf{a}_i} p(\mathbf{a}_i)\nabla \log p(\mathbf{x}_t|\mathbf{a}_i). \tag{8}$$

This reveals that the unconditional score is a convex combination of conditional scores. Now, consider a binary feature scenario where $p(\mathbf{a}_1) \ll p(\mathbf{a}_2)$. The unconditional score becomes dominated by the conditional score of $\mathbf{a}_2$, approximating $\nabla \log p(\mathbf{x}_t) \approx \nabla \log p(\mathbf{x}_t|\mathbf{a}_2)$. In general, this results in biased generation where the score function favors specific distribution modes rather than exploring the complete modal space. We hypothesize this as a primary source of inherent bias in DMs. To verify this hypothesis, we conduct a controlled experiment using a mixture of Gaussian (MoG) distributions as illustrated in Fig. 2. We take an ideal true distribution with balanced mixture weights (50:50 split between modes), but deliberately introduce sampling bias during training (10:90 ratio), mimicking real-world scenarios of underrepresented classes. After training a DM on this data, we observe a biased generation towards one mode, further after estimating the weights as in Eq. 8, we see that the weights are skewed towards one mode ($0.12 \ll 0.88$) verifying our hypothesis[2].

From Eq. 8, we observe that attribute influence in the score function follows the proportion of $p(\mathbf{a}_i)$ learned by pre-trained DMs. However, our goal is to align this with user-provided $\mathbf{p^a_{ref}}$ proportions. We therefore decompose the reference distribution-based debiasing into two components: (a) sample tagging to maintain attributes in $\mathbf{p^a_{ref}}$ proportions, and (b) score guidance which, using Eq.4, equivalently guides $\hat{\mathbf{x}}_{0|t}$. This is illustrated in Fig. 1

Building upon these theoretical foundations, we present debiasing approaches for DMs using two modalities: text and exemplar images. For text-based debiasing, we leverage pre-trained multi-modal models such as CLIP to guide the score functions towards desired modes of the distribution. For exemplar-based debiasing, we introduce a novel approach using DDIM inversion to achieve targeted score guidance. We elaborate on both methodologies in the subsequent sections.

**Remark on Attribute Entanglement:** In real-world datasets, attributes are frequently entangled (e.g., specific genders being statistically correlated with certain professions). It is important to clarify that Equation 8 does not assume attributes are disentangled. Rather, it represents an exact marginal projection over a specific target attribute $A = \{\mathbf{a}_1, \mathbf{a}_2\}$. Any other highly entangled attributes $B \in \{\mathbf{b}_j\}$ are implicitly marginalized out within the conditional score term. Mathematically, the conditional score naturally absorbs this entanglement:

$$\nabla \log p(\mathbf{x}_t|\mathbf{a}_i) = \sum_j p_{\text{train}}(\mathbf{b}_j|\mathbf{x}_t, \mathbf{a}_i)\nabla \log p(\mathbf{x}_t|\mathbf{a}_i, \mathbf{b}_j) \tag{9}$$

---

[1]Specifically, $\nabla \log p(\mathbf{x}_t) = -\frac{1}{\sqrt{1 - \bar{\alpha}_t}}\epsilon_\theta(\mathbf{x}_t, t)$

[2]The origin of the prior probability weights ($p(\mathbf{a}_i)$) is unconstrained (they could emerge as a result of artifact in model's training or due to underlying data distribution). Our toy experiment with a MoG was designed to isolate and demonstrate how these weights manifest as sampling bias. We provide the code to reproduce this result in Supplementary.

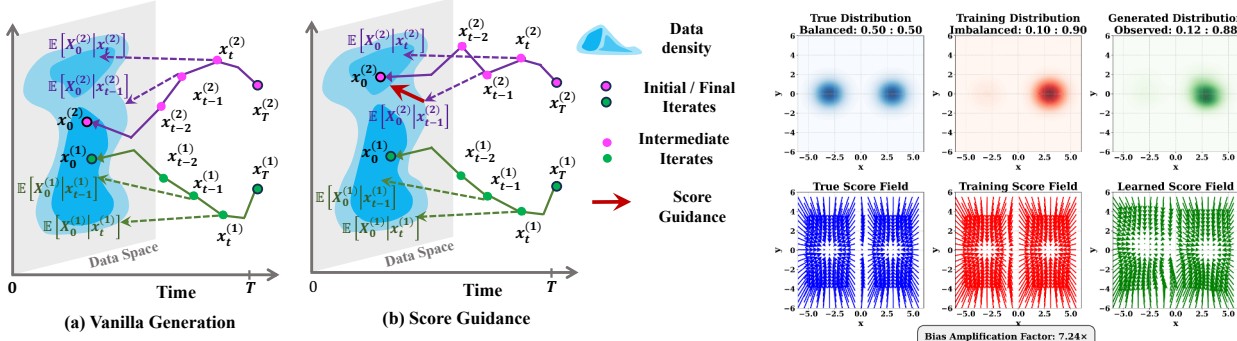

Figure 1: (a) Standard generation, where one mode dominates, causing under-representation of others (b) Proposed method, which tags and guides samples.

Figure 2: A demonstration of bias in Diffusion Models. The output bias confirms that the model learns skewed score weights from the data.

Consequently, guiding the generation process solely on attribute $A$ will inadvertently leave the samples biased with respect to the entangled attribute $B$, as the SDE is pulled by the skewed training prior $p_{\text{train}}(b_j|a_i)$.

To resolve this, our framework natively supports simultaneous multi-attribute guidance (detailed in Section 4.1.2). We break this entanglement through two mechanisms: (1) *Recursive Tagging*: We construct a statistically independent joint reference $p_{\text{ref}}(\mathbf{a}_i, \mathbf{b}_j) = p_{\text{ref}}(\mathbf{a}_i)p_{\text{ref}}(\mathbf{b}_j)$ and recursively tag the batch to exactly match these proportions, bypassing the biased $p_{\text{train}}(\mathbf{b}_j|\mathbf{a}_i)$. (2) *Alternating Projections*: By performing sequential score updates for $A$ and then $B$, we effectively apply Projected Gradient Descent on a composite potential $U_{\text{joint}}(\mathbf{x}) = U_A(\mathbf{x}) + U_B(\mathbf{x})$. By the theory of alternating projections, the predicted clean latent state $\hat{\mathbf{x}}_{0|t}$ converges to the intersection of the attribute manifolds ($\mathcal{B}_{\mathbf{a}_i} \cap \mathcal{B}_{\mathbf{b}_j}$), effectively disentangling the attributes during inference.

### 3.3 Text-based Debiasing

For text-based debiasing of DMs, we utilize textual descriptions of the attributes under consideration. For instance, to address gender bias, we might use descriptions like 'a male person' and 'a female person' corresponding to the classes '*male*' and '*female*' respectively. More formally, given an attribute $\mathbf{a}$ with $n$ distinct classes $\{\mathbf{a}_i\}_{i=1}^n$, we associate each class with a corresponding textual description $\{\mathbf{t}_i\}_{i=1}^n$.

During inference, we incorporate a user-specified reference distribution $\mathbf{p}_{\text{ref}}^{\mathbf{a}} = \{p_i\}_{i=1}^n$, where $p_i$ represents the desired proportion for attribute class $\mathbf{a}_i$. The debiasing process occurs during a selected time window $\mathcal{T} = \{t_{\text{start}} = t_s, t_{s-1}, \cdots, t_e = t_{\text{end}}\}$ within the reverse process that denoises samples from timestep $T$ to 0. At time $t_s$, we perform sample tagging by computing the cosine similarity between CLIP embeddings of each text description $\mathbf{t}_i$ and the corresponding image $\hat{\mathbf{x}}_{0|t_s}$[3].

To illustrate the process, consider a binary attribute case with batch size $B$. Our goal is to generate $p_1 B$ samples from the first class and $p_2 B$ samples from the second class. At timestep $t_s$, we compute the CLIP similarity between $\mathbf{t}_1$ and each sample in $\{\hat{\mathbf{x}}_{0|t_s}^{(i)}\}_{i=1}^B$. The samples are then tagged by assigning $\mathbf{a}_1$ to the top $p_1 B$ samples with highest cosine similarity, while the remaining samples are tagged with $\mathbf{a}_2$. This methodology naturally extends to scenarios with multiple attribute classes by iteratively applying the tagging process according to the desired proportions in $\mathbf{p}_{\text{ref}}^{\mathbf{a}}$.

Following the tagging process, our objective is to guide the score of each sample towards its corresponding attribute mode. This guidance is achieved by utilizing the gradient of CLIP similarity to update $\hat{\mathbf{x}}_{0|t_s}^{(i)}$. However, direct guidance in the high-dimensional image space poses significant challenges, as previously observed by Parihar et al. (2024). To address this limitation, we adopt an approach similar to Parihar et al. (2024) by operating in the bottleneck $\mathbf{H}$-space of the UNet-based denoiser rather than the image space

---

[3]We use 'CLIP similarity' as shorthand for the cosine similarity between CLIP embeddings of a text-image pair.

directly. The guidance update can be formally expressed as:

$$\mathbf{h}^{(i)} = \mathbf{h}^{(i)} - \gamma \nabla_{\mathbf{h}^{(i)}} \left( 1 - \frac{\mathsf{clip}(\mathbf{t}^{(i)}) \cdot \mathsf{clip}(\hat{\mathbf{x}}_{0|t}^{(i)})}{|\mathsf{clip}(\mathbf{t}^{(i)})||\mathsf{clip}(\hat{\mathbf{x}}_{0|t}^{(i)})|} \right) \tag{10}$$

where $\gamma$ is the guidance strength and $\mathbf{t}^{(i)}$ represents the text assigned to the $i$th sample during the tagging process. This update is applied $M$ times to the $\mathbf{H}$-space representation for each timestep within $\mathcal{T}$. Specifically, $\hat{\mathbf{x}}_{0|t}^{(i)}$ is related to $\epsilon_\theta(\mathbf{x}_t^{(i)}, t)$ via Eq. 5. Following Kwon et al. (2022), considering the UNet-type architecture, we can write the denoiser network as $\epsilon_\theta(\mathbf{x}_t^{(i)}, t) = D_\theta \circ E_\theta(\mathbf{x}_t^{(i)}, t)$ where $D_\theta(\cdot)$ and $E_\theta(\cdot)$ are the decoder and encoder part of the network. The $\mathbf{H}$-space representation can be denoted as: $\mathbf{h}^{(i)} = E_\theta(\mathbf{x}_t^{(i)}, t)$. As shown in Kwon et al. (2022), $\mathbf{H}$-space is more semantically meaningful and easier to drive the generation in diffusion models, we leverage this property for our purpose. $\nabla_{h^{(i)}}$ represents the gradient with respect to this representation.

While prior work notes that CLIP guidance requires robust embeddings for noisy data (Parihar et al., 2024), our method holds an advantage by operating on the predicted clean sample, $\hat{\mathbf{x}}_{0|t}$, instead of the noisy input $\mathbf{x}_t$. We further enhance effectiveness by optimizing the guidance time window, $\mathcal{T}$. This leverages the established finding that features emerge at specific intervals in the diffusion process (Dieleman, 2023; Meng et al., 2022; Li & Chen, 2024; Choi et al., 2022; Raya & Ambrogioni, 2024; Georgiev et al., 2023), allowing us to apply guidance when features are most distinct. We also discuss and provide results for other robustness related points in Section C.1 of Supplementary.

### 3.4  Exemplar-based Debiasing

In exemplar-based debiasing, we utilize a small set of representative images for each attribute class[4]. For instance, in addressing gender bias, we would incorporate exemplar images representing both '*male*' and '*female*' categories. More formally, each attribute class $\mathbf{a}_i$ is associated with a set of $k$ exemplar images, denoted as $\{\mathbf{e}^{(i,j)}\}_{j=1}^k$.

We begin by processing these exemplar images through DDIM Inversion (Song et al., 2021; Mokady et al., 2023) to obtain their corresponding intermediate latent representations in the pretrained DM's space:

$$\left\{ \bar{\mathbf{e}}_t^{(i,j)}, \ \bar{\epsilon}_t^{(i,j)} \right\}_{t=1}^T = \mathsf{DDIM\text{-}Inversion}(\mathbf{e}^{(i,j)}) \tag{11}$$

$$\bar{\mathbf{e}}_{0|t}^{(i,j)} \triangleq \frac{\bar{\mathbf{e}}_t^{(i,j)} - \sqrt{1 - \bar{\alpha}_t} \bar{\epsilon}_t^{(i,j)}}{\sqrt{\bar{\alpha}_t}} \quad \forall \ t \in [1, T] \tag{12}$$

where we have used Eq. 5 to obtain estimates of final denoised sample using intermediate noisy samples obtained via DDIM inversion. Next, for each attribute class, we take the average of these estimate at each time step across all exemplar to obtain 'anchor' points as follows:

$$\bar{\mathbf{e}}_{0|t}^{(i)} = \frac{1}{k} \sum_j \bar{\mathbf{e}}_{0|t}^{(i,j)} \quad \forall \ i. \tag{13}$$

These anchor points, $\bar{\mathbf{e}}_{0|t}^{(i)}$, can be interpreted as estimates of mean of the conditional expectation $\mathbb{E}[\mathbf{X}_0 \mid \mathbf{X}_t, \mathbf{Y} = \mathbf{a}_i]$[5]. By the Law of Large Numbers, these estimates converge to the true conditional expectation as the number of exemplar samples - $n$ increases. We provide results to investigate the effect of number of exemplar samples in Appendix. Alternatively, they can be thought of as prototypical denoised samples embodying attribute $\mathbf{a}_i$. With these $n$ anchor points corresponding to the $n$ attribute classes, we can effectively guide the generation process to incorporate the desired attribute characteristics in the generated samples.

---

[4]In our experiments, we use only eight exemplar images from each class.
[5]Note that $\mathbb{E}[\mathbf{X}_0 \mid \mathbf{X}_t, \mathbf{Y} = \mathbf{a}_i]$ is a random variable.

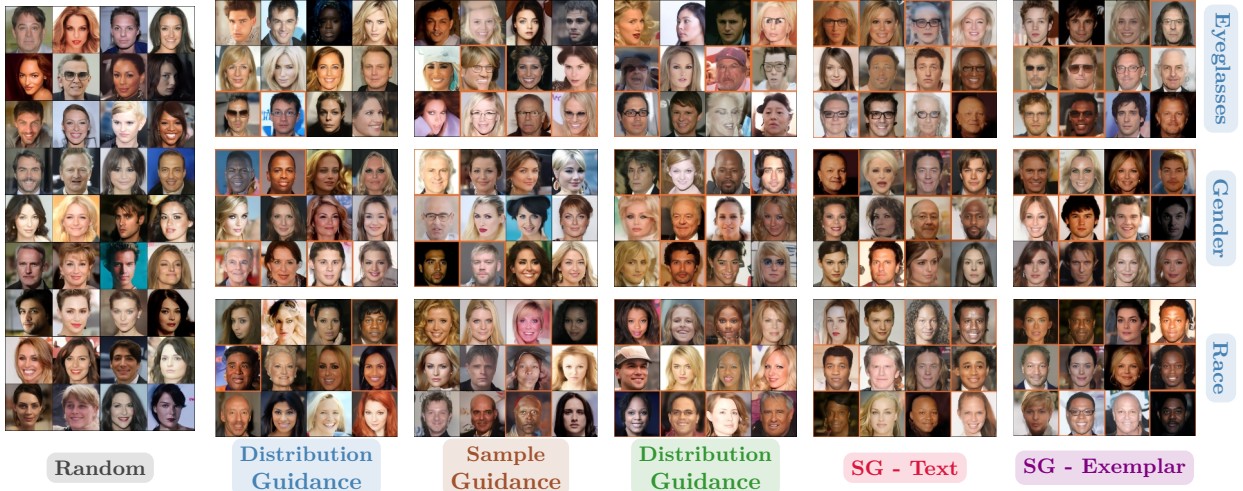

Figure 3: Visualization of balanced generation on 'eyeglasses' and 'gender' using different baselines and our method. The samples with eyeglasses and male gender are shown in orange colored border.

Following the computation of anchor points, we define a time window $\mathcal{T} = \{t_{\text{start}} = t_s, t_{s-1}, \cdots, t_e = t_{\text{end}}\}$ for guidance, similar to our text-based debiasing approach. Within this window, we implement sample tagging based on the $\ell_2$-distance from the anchor points.

To illustrate this process, again consider a binary attribute case where our objective is to generate a batch of $B$ samples, with $p_1 B$ samples from the first class and $p_2 B$ samples from the second class. At timestep $t_s$, we compute the $\ell_2$-distance between each sample and the first class anchor point $\bar{\mathbf{e}}_{0|t_s}^{(1)}$ (derived from exemplar images). The tagging process then assigns attribute $\mathbf{a}_1$ to the $p_1 B$ samples closest to this anchor point, while the remaining samples are designated as $\mathbf{a}_2$.

Following the tagging process, our goal is to guide each sample toward its corresponding anchor point. We achieve this through a simple geometric approach that updates $\hat{\mathbf{x}}_{0|t}^{(i)}$ to ensure it remains within an $r$-radius ball centered at $\bar{\mathbf{e}}_{0|t}^{(j)}$ (where the $i$th sample has been tagged with attribute $\mathbf{a}_j$). This guidance is implemented through the following update equation:

$$\hat{\mathbf{x}}_{0|t}^{(i)} = \hat{\mathbf{x}}_{0|t}^{(i)} - \left(\hat{\mathbf{x}}_{0|t}^{(i)} - \bar{\mathbf{e}}_{0|t}^{(j)}\right)\left(1 - \frac{r}{\|\hat{\mathbf{x}}_{0|t}^{(i)} - \bar{\mathbf{e}}_{0|t}^{(j)}\|}\right) \tag{14}$$

This update is applied for all timesteps $t \in \mathcal{T}$. In practice, we update the $\mathbf{H}$-space vectors similar to text-based debiasing. Refer to supplementary for details. The formulation ensures that $\hat{\mathbf{x}}_{0|t}^{(i)}$ maintains proximity to its designated anchor point $\bar{\mathbf{e}}_{0|t}^{(j)}$, providing an elegant and effective mechanism for incorporating desired attributes into the generated samples.

Using this guidance mechanism, we can formally guarantee that the final denoised sample will closely approximate the mean of the conditional distribution given the target attribute. We formalize this as follows:

**Theorem 3.1** (Informal). *Under the guidance mechanism defined in Eq. 14, the following bound holds almost surely:*

$$\|\mathbb{E}[\mathbf{X}_0] - \mathbb{E}[\mathbf{X} \mid \mathbf{Y} = \mathbf{a}]\| \leq r \tag{15}$$

We refer to Supplementary (Section A) for a formal statement and proof. Hence, we establish a theoretical bound on how far the generated samples can deviate from the desired target. The parameter $r$ effectively controls the trade-off between diversity and accuracy—smaller values of $r$ produce samples closer to the

conditional expectation but with potentially less diversity, while larger values permit more variation in the generated outputs while still maintaining statistical fidelity to the conditioning information.

## 4 Experiments

In this section, we evaluate our proposed method - Score Guidance (SG) - on both unconditional and conditional diffusion models. For unconditional generation, we conduct experiments using the P2 model (Choi et al., 2022) trained on the CelebA-HQ dataset. For conditional generation, we employ the Stable Diffusion v1.5 model (Rombach et al., 2022). Our comprehensive evaluation demonstrates that our method consistently outperforms existing baselines across all metrics.

**Evaluation Metrics**: Following the approach of Parihar et al. (2024), we employ two primary metrics. The first metric, Fairness Discrepancy (FD) (Choi et al., 2020), quantifies the bias in generated samples by measuring the deviation from uniform distribution across concerned attribute classes. Specifically, FD computes the difference between a uniform vector $\bar{p}$ and the average softmax activations obtained from a pre-trained high-accuracy classifier $\mathcal{C}_{\mathbf{a}}$:

$$\text{FD} = \|\bar{p} - \mathbb{E}_{\mathbf{x} \sim p_\theta(\mathbf{x})}[\mathcal{C}_{\mathbf{a}}(\mathbf{x})]\|_2 \tag{16}$$

where $\bar{p}$ represents a uniform vector whose dimension matches the number of classes in the attribute space. The second metric, Fréchet Inception Distance (FID) (Heusel et al., 2017), evaluates the quality and diversity of generated samples. We follow the convention of presenting the best and second best performance in **bold** and underlined text respectively.

**Baselines:** For unconditional generation, we evaluate our method against four major baseline approaches. The first baseline is Universal Guidance (Bansal et al., 2023), which follows a score-guidance paradigm using a classifier on $\hat{\mathbf{x}}_{0|t}$ for guidance. While they train a classifier on clean images and utilize its gradients for guidance, such classifiers must be robust to $\hat{\mathbf{x}}_{0|t}$, particularly during initial timesteps where the estimated $\mathbf{x}_0$ is less accurate, potentially leading to imprecise gradient estimation and guidance. In contrast, our method applies guidance only within a specific time window $\mathcal{T}$, determined through extensive prior research (Meng et al., 2022; Li & Chen, 2024; Choi et al., 2022; Raya & Ambrogioni, 2024). Following Parihar et al. (2024), we examine two variants of Universal Guidance: one with a classifier trained on 2k **H**-space samples and another trained on the complete CelebA-HQ dataset of 30k samples.

The second baseline, Latent Editing (Kwon et al., 2022), focuses on editing the **H**-space vectors toward particular attributes. The third baseline, **H**-space Guidance (Parihar et al., 2024), introduces a lightweight classifier trained on **H**-space for guidance purposes. They propose two variants: one that updates the **H**-space vectors directly and another that matches the classifier's softmax predictions with a provided reference distribution. The fourth baseline, Magnet (Humayun et al., 2021), proposes uniform sampling from the image manifold, though it is based on the StyleGAN2 model, and we report results accordingly.

We evaluate several baselines for conditional generation in Stable Diffusion. ITI-Gen(Zhang et al., 2023a) learns and combines debiased prompt embeddings during inference. Fair Diffusion (Friedrich et al., 2023) uses a dictionary to identify biased concepts and adds scaled attribute expressions to prompts. Fair Mapping (Li et al., 2024) balances the attribute embedding distance from the class embedding, while ADFT (Shen et al., 2024) finetunes text tokens to align classifier logits with a reference distribution. We also include the editing methods UCE (Gandikota et al., 2024) and TIME (Orgad et al., 2023), and our previously described H-space Guidance.

Additionally, we establish a baseline performance measure through random sampling from the diffusion models under consideration. We compare our method, using both - Text (SG-Text) and Exemplar images (SG-Exemplar), against all the baselines. We refer to supplementary for more results (Section C.3, C), visualizations(Section C.3, D), ablation studies (Section F) and code implementation (Section B).

Table 2: Results for balanced generation on binary class attributes

| Method | Gender | | Race | | Eyeglasses | |
|---|---|---|---|---|---|---|
| | FD ($\downarrow$) | FID ($\downarrow$) | FD ($\downarrow$) | FID ($\downarrow$) | FD ($\downarrow$) | FID ($\downarrow$) |
| *StyleGAN2 Models* | | | | | | |
| **StyleGAN2 - Random Sampling** | 0.307 | 112.28 | 0.463 | 123.97 | 0.276 | 117.83 |
| **StyleGAN2 - Magnet** (Humayun et al., 2021) | 0.267 | 91.15 | 0.454 | 97.05 | 0.281 | 106.55 |
| *Baseline Methods* | | | | | | |
| **Random Sampling** | 0.178 | 54.59 | 0.334 | 60.01 | 0.251 | 75.21 |
| **Universal Guidance (2k)** (Bansal et al., 2023) | 0.193 | 52.10 | 0.377 | 93.42 | 0.189 | 64.55 |
| **Universal Guidance (30k)** (Bansal et al., 2023) | 0.127 | 48.94 | 0.326 | 58.52 | 0.051 | 78.57 |
| **Latent Editing** (Kwon et al., 2022) | **0.001** | 37.40 | 0.214 | 42.69 | 0.330 | 75.04 |
| **H-Sample Guidance** (Parihar et al., 2024) | 0.113 | 51.46 | 0.184 | 56.53 | 0.118 | 57.63 |
| **H-Distribution Guidance** (Parihar et al., 2024) | 0.049 | 50.27 | 0.113 | 52.38 | 0.014 | 51.78 |
| *Our Methods* | | | | | | |
| **SG - Text (ours)** | 0.022 | 35.24 | 0.093 | 43.08 | 0.116 | 55.24 |
| **SG - Exemplar (ours)** | **0.001** | **34.61** | **0.024** | **39.77** | **0.012** | **48.80** |

## 4.1 Main Results

### 4.1.1 Binary Class Attributes

We evaluate our method on the unconditional P2 diffusion model (Choi et al., 2022) across three binary class attributes: Gender (male and female), Race (black and white), and Eyeglasses (wearing and not wearing). Our objective is to debias the model to generate an equal proportion of samples for each class ($\mathbf{p^a_{ref}} = [0.5, 0.5]$). For the SG-Text approach, we employ descriptive texts such as '*a male person*' and '*a female person*' for score guidance. In the SG-Exemplar approach, we utilize eight exemplar images from each class. Detailed hyperparameters and implementation specifications are provided in the Supplementary material Section B.

The qualitative results of our proposed method on eyeglasses and gender are presented in Fig. 3. Our experiments reveal several limitations in existing approaches. Universal guidance and Sample guidance methods often fail to maintain the desired reference distribution in their generated samples. Distribution guidance often generates samples with noticeable artifacts, moreover, we observed that it exhibits performance degradation with smaller batch sizes, likely due to its reliance on an estimated surrogate distribution $\mathbf{p^a_\theta}$ for guidance, which requires large batch sizes for accurate estimation. In contrast, our approach of explicitly tagging samples and guiding them towards desired modes circumvents these limitations, resulting in more reliable and consistent performance.

The quantitative results for debiasing are presented in Table 2. Our proposed SG-Exemplar (abbreviated as SG (E)) demonstrates superior performance across both evaluation metrics compared to all other methods, indicating its ability to generate debiased samples while preserving generation quality. In gender debiasing, SG (E) and Latent Editing achieve comparable FD scores, with SG (E) showing marginally better FID performance. SG-Text (abbreviated as SG (T)) emerges as the second-best performer across both metrics. For race debiasing, SG (E) and SG (T) achieve the best and second-best FD scores, showing improvements of 8.9% and 2% respectively over **H**-distribution guidance. SG (E) also achieves the best FID score, demonstrating an improvement of 2.92 points over the second-best performer. In the eyeglasses attribute, SG (E) again outperforms other methods on both metrics, with **H**-distribution guidance ranking second. Specifically, SG (E) surpasses **H**-distribution guidance by a margin of 0.2% in FD and approximately 3 points in FID. These comprehensive results demonstrate that SG (E) consistently outperforms existing methods across all metrics, while SG (T) delivers performance that is either comparable to or better than other baseline approaches.

We also provide qualitative results on arbitrary reference distribution in Fig. 4 and quantitative results in Table 3. We again observe that SG (E) gives the best FD while maintaining marginally better FID compared to distribution guidance.

Table 3: Results on imbalanced generation

| Method | 0.2F - 0.8M | | 0.1W - 0.9B | |
|---|---|---|---|---|
| | FD ($\downarrow$) | FID ($\downarrow$) | FD ($\downarrow$) | FID ($\downarrow$) |
| Random Sampling | 0.478 | 72.26 | 0.734 | 77.63 |
| H-Distribution Guidance (Parihar et al., 2024) | 0.168 | 51.65 | 0.325 | 53.80 |
| SG - Text (ours) | 0.130 | 56.26 | 0.307 | 58.11 |
| SG - Exemplar (ours) | **0.093** | **50.61** | **0.251** | **51.08** |

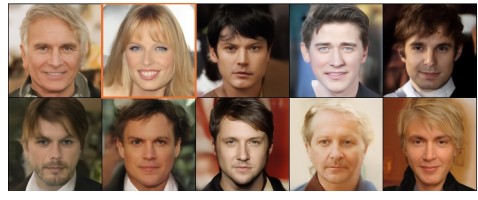
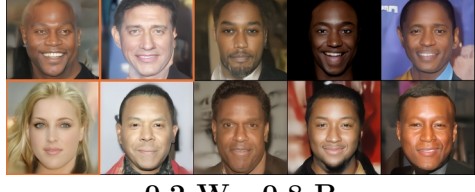

**0.1 F - 0.9 M**          **0.2 W - 0.8 B**

Figure 4: Visual results for skewed reference distributions using SG - Exemplar. The minority class is denoted orange color. (F = female, M = male, W = white, B = black).

Table 4: Results for balanced generation on multiple attributes

| Method | Gender + Race | | Eyeglasses + Race | | Gender + Eyeglasses | |
|---|---|---|---|---|---|---|
| | FD ($\downarrow$) | FID ($\downarrow$) | FD ($\downarrow$) | FID ($\downarrow$) | FD ($\downarrow$) | FID ($\downarrow$) |
| Random Sampling | 0.256 | 60.68 | 0.292 | 89.14 | 0.214 | 70.97 |
| Latent Editing (Kwon et al., 2022) | 0.124 | 64.84 | 0.219 | 90.63 | 0.230 | 74.93 |
| Universal Guidance (2k) (Bansal et al., 2023) | 0.283 | 71.84 | 0.264 | 91.54 | 0.157 | 80.57 |
| H-Sample Guidance (Parihar et al., 2024) | 0.241 | 59.78 | 0.135 | 67.87 | 0.079 | 52.03 |
| H-Distribution Guidance (Parihar et al., 2024) | 0.075 | 49.91 | **0.101** | 57.46 | 0.057 | 52.03 |
| SG - Text (ours) | 0.173 | 50.81 | 0.132 | 54.90 | 0.062 | 49.33 |
| SG - Exemplar (ours) | **0.028** | **46.83** | 0.125 | **51.30** | **0.051** | **45.42** |

### 4.1.2 Multiple Attributes

We extend our evaluation to simultaneous debiasing of multiple attributes. In this scenario, we consider two attributes, $\mathbf{a}_1$ and $\mathbf{a}_2$, with their corresponding reference distributions, $\mathbf{p}_{\mathbf{ref}}^{\mathbf{a}_1}$ and $\mathbf{p}_{\mathbf{ref}}^{\mathbf{a}_2}$. The objective is to generate samples that conform to both reference distributions simultaneously. For evaluation, we employ a balanced reference distribution that aligns with our single-attribute debiasing experiments. The implementation involves sequential application of our method across multiple attributes during the guidance phase. Taking the example of simultaneous gender and race debiasing, we first compute and apply updates for gender followed by updates for race attributes. This process effectively mirrors the principles of projected gradient descent in optimization theory. The results of this multi-attribute debiasing are presented in Table 4.

Our observation shows that SG (E) demonstrates superior performance compared to baseline methods in the majority of cases. In the joint debiasing of gender and race attributes, SG (E) achieves a 4.7% improvement in FD over **H**-distribution guidance while simultaneously maintaining better generation quality with an FID improvement of 3.08 points. Similar performance advantages are observed in the gender and eyeglasses combination, where SG (E) shows marginal improvement in FD while achieving a substantial FID improvement of 6.61 points. In the case of eyeglasses and race attributes, while **H**-distribution guidance achieves the best FD performance, surpassing SG (E) by 2.4%, this comes at the cost of generation quality, with an FID score 6.16 points worse than our method. Further, results using skewed distribution on multiple attributes can be found in Supplementary Section C.3.

Table 5: Results for balanced generation on multi-class attributes

| Method | Age (3 classes) | | Race (4 classes) | |
|---|---|---|---|---|
| | FD ($\downarrow$) | FID ($\downarrow$) | FD ($\downarrow$) | FID ($\downarrow$) |
| **Random Sampling** | 0.256 | 60.68 | 0.292 | 89.14 |
| **H-Sample Guidance** (Parihar et al., 2024) | 0.124 | 64.84 | 0.219 | 90.63 |
| **H-Distribution Guidance** (Parihar et al., 2024) | 0.283 | 71.84 | 0.264 | 91.54 |
| **SG - Text (ours)** | 0.119 | 74.81 | 0.065 | 95.01 |
| **SG - Exemplar (ours)** | **0.047** | **58.35** | **0.010** | **88.22** |

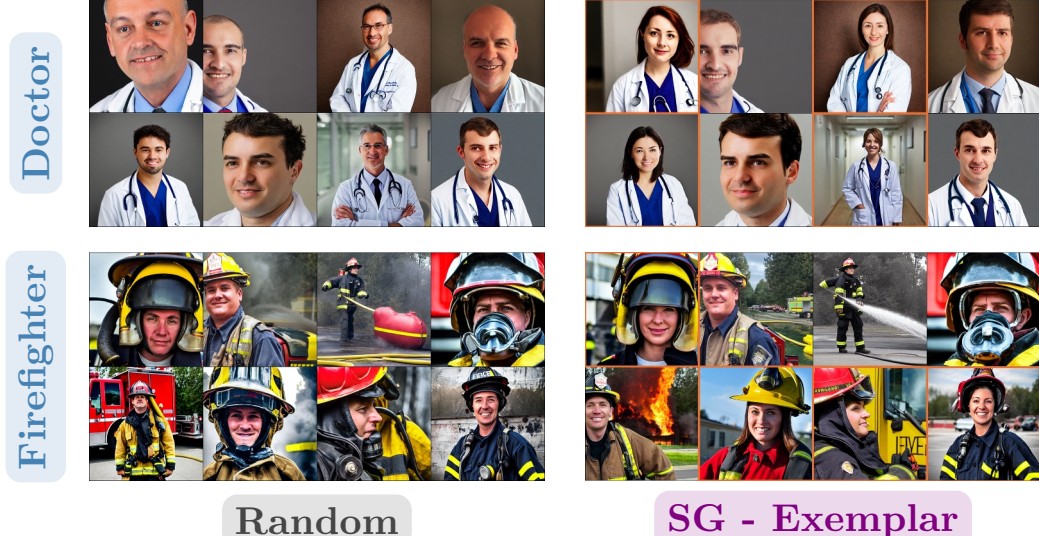

Figure 5: Visualizations of 'gender-balanced' samples for different profession - doctor and firefighter - from Stable Diffusion using SG (E). More visualizations are provided in Supplementary.

### 4.1.3 Multi-Class Attributes

We extend our evaluation to multi-class attributes, where the objective is to achieve balanced generation across multiple attribute classes. Following Parihar et al. (2024), we examine two attributes: Age, comprising three classes (young, adult, and old), and Race, containing four classes (black, brown, asian, and white). The quantitative results are presented in Table 5.

The results demonstrate that SG (E) consistently outperforms existing methods across all metrics for both attributes, with SG (T) achieving the second-best performance in terms of FD. A notable observation is that while baseline methods struggle to maintain FID scores comparable to random sampling, SG (E) actually improves the FID scores in all cases, achieving enhancements of 2.33 and 0.92 points for age and race attributes respectively. We attribute this superior performance to our method's non-gradient based guidance approach in exemplar-based debiasing. Furthermore, our method achieves substantial improvements in FD compared to previous state-of-the-art approaches, with SG (E) demonstrating FD improvements of 7.7% and 20.9% over **H**-space guidance methods. These results indicate that our method successfully extends the strong performance observed in binary class attributes to the multi-class attribute scenario, distinguishing itself from previous debiasing approaches. We provide results for even more complex skewed generation in multi-class setting in Supplementary which shows the robustness of our method to complicated debiasing scenarios.

### 4.2 Debiasing Conditional Text-to-Image Diffusion Models

We evaluate our method on conditional text-to-image generation using the Stable Diffusion (SD) v1.5 model (Rombach et al., 2022). Additional results for SDv2.0 (Stability AI, 2022) and SDXL (Podell et al.,

Table 6: Results for balanced generation on Stable Diffusion

| Method | Gender | | Doctor | | Firefighter | |
|---|---|---|---|---|---|---|
| | FD ($\downarrow$) | FID ($\downarrow$) | FD ($\downarrow$) | FID ($\downarrow$) | FD ($\downarrow$) | FID ($\downarrow$) |
| *Editing Methods* | | | | | | |
| **UCE** (Gandikota et al., 2024) | 0.178 | 74.33 | 0.118 | 73.02 | 0.216 | 72.55 |
| **TIME** (Orgad et al., 2023) | 0.127 | 68.29 | 0.085 | 70.09 | 0.104 | 69.67 |
| *Debiasing Methods* | | | | | | |
| **Random Sampling** | 0.317 | 72.37 | 0.355 | 70.11 | 0.235 | 71.86 |
| **ITI-Gen** (Zhang et al., 2023a) | 0.049 | 64.79 | 0.072 | 67.81 | 0.184 | 70.12 |
| **Fair Mapping** (Li et al., 2024) | 0.233 | 72.08 | 0.049 | 69.17 | 0.090 | 70.01 |
| **Fair Diffusion** (Friedrich et al., 2023) | 0.227 | 71.22 | 0.035 | 74.37 | **0.036** | 68.33 |
| **ADFT** (Shen et al., 2024) | 0.116 | 63.82 | 0.015 | 70.41 | 0.059 | 67.55 |
| **H-Sample Guidance** (Parihar et al., 2024) | 0.026 | 70.96 | 0.021 | 68.43 | 0.097 | 70.42 |
| **H-Distribution Guidance** (Parihar et al., 2024) | 0.024 | 70.69 | 0.015 | 67.36 | 0.093 | 69.41 |
| *Our Methods* | | | | | | |
| **SG - Text (ours)** | 0.027 | 71.33 | 0.025 | 67.36 | 0.072 | 68.73 |
| **SG - Exemplar (ours)** | **0.011** | **60.72** | **0.001** | **66.02** | 0.056 | **66.58** |

2023) are presented in Section E of the Supplementary. Prior research (Zhang et al., 2023a) has shown that SD models exhibit gender biases in profession-related generations, particularly associating certain professions with specific genders. We focus on two such professions - 'doctors' and 'firefighters' - which have been documented to show significant gender bias in SD models (Zhang et al., 2023a; Friedrich et al., 2023).

For SG (T), we implement the same approach used in unconditional diffusion models, utilizing attribute text for guidance. In implementing SG (E), we first collect exemplar images through the conditional diffusion model using specific prompts. For instance, to address gender bias in doctor-related generations, we generate exemplar samples using prompts such as 'a photo of a male doctor' and 'a photo of a female doctor' to obtain male and female exemplar images respectively. We then apply our debiasing methodology as previously described.

Qualitative results are shown in Fig. 5, with additional results in the Supplementary. The quantitative results in Table 6 demonstrate that SG (E) achieves superior performance in most scenarios. Specifically, SG (E) improves FD over **H**-space guidance by 1.4% for doctors and 3.7% for firefighters. While Fair Diffusion has the best FD score for firefighters, SG (E) offers a favorable trade-off with a better FID score. These results validate our method's effectiveness in addressing biases within conditional diffusion models.

## 5 Conclusion

In this work, we present a novel training-free, inference-time approach for mitigating biases in diffusion models (DMs). We begin by providing a simple mathematical explanation to highlight the inherent biases present in DMs. To address these biases, we introduce a solution based on 'score-guidance,' which can be implemented through two distinct modalities: text and exemplar images. Importantly, our framework natively accommodates multi-attribute sequential guidance, providing a structured mathematical mechanism to counteract real-world attribute entanglement without requiring dataset curation or model fine-tuning. We demonstrate the effectiveness of our method on both unconditional and conditional text-to-image DMs, including Stable Diffusion. Extensive experiments reveal that our approach consistently outperforms existing state-of-the-art baseline methods, underscoring its capability to effectively reduce biases in pre-trained DMs. These findings suggest that the proposed method is a robust and efficient solution for debiasing generative models without the need for additional training.

## Broader Impact Statement and Limitations

The proposed training-free, inference-time debiasing method offers a critical advancement in addressing the ethical challenges posed by biases in diffusion models (DMs). By correcting unintended biases during inference,

the approach ensures outputs align with equitable standards across demographic and societal groups. This promotes balanced representation in AI-generated content, reducing the risks of harmful stereotypes and systemic inequities. Furthermore, the method's simplicity—requiring no retraining or auxiliary classifier training—lowers the barriers to adoption, enabling organizations to deploy ethically aligned DMs in sensitive domains without resource-intensive overhauls. However, we recognize that defining fairness for generative models is inherently context-dependent and heavily influenced by external socio-technical factors. Our algorithmic formulation relies on achieving statistical parity via user-provided reference distributions, which captures only one mathematical dimension of fairness. Deploying truly equitable generative models requires embedding such debiasing tools within a broader, human-in-the-loop pipeline that actively evaluates the downstream societal context and cultural nuances of the generated media. Finally, from a technical perspective, our method introduces certain operational trade-offs. The SG-Text approach requires computing gradients through both the UNet denoiser and the CLIP text encoder at each guided timestep, which naturally increases inference latency and memory overhead compared to standard generation (see Supplementary Section C.2 for a detailed analysis). Additionally, our SG-Exemplar method—which computes anchor points by averaging the inverted latents of exemplar images—inherently assumes that the class-conditioned latent distribution is uni-modal. While this assumption proves highly effective in practice, highly complex or disjoint attribute topologies may require more careful exemplar selection.

**Acknowledgements**

This work was supported (in part for setting up the GPU compute) by the Indian Institute of Science through a start-up grant. Piyush is supported by Government of India via Prime Minister's Research Fellowship. Prathosh is supported by Infosys Foundation Young investigator award.

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

# A  Theoretical Analysis

## A.1  Preliminaries

We begin by introducing the mathematical foundations necessary for our theoretical analysis. The standard form of a Stochastic Differential Equation (SDE) (Øksendal, 2003) is given by:

$$d\mathbf{X}_t = f(\mathbf{X}_t, t)dt + g(t)d\mathbf{B}_t \tag{17}$$

where $f(\cdot, \cdot)$ and $g(\cdot)$ represent the drift and diffusion terms, respectively.

For generative modeling, we are particularly interested in the reverse-time process. According to Anderson (1982), the corresponding reverse SDE is formulated as:

$$d\mathbf{X}_t = \left[ f(\mathbf{X}_t, t) - g(t)^2 \nabla_{\mathbf{x}} \log p_t(\mathbf{x}) \right] dt + g(t)d\mathbf{B}_t \tag{18}$$

where $p_t(\mathbf{x})$ denotes the marginal distribution of the random variable at time $t$ in the reverse process. The term $\nabla_{\mathbf{x}} \log p_t(\mathbf{x})$ is known as the score function.

In practice, this score function is approximated using a learned parametric model $s_\theta(\mathbf{x}_t)$, allowing us to express the reverse SDE as (Song & Ermon, 2019; Song et al., 2021):

$$d\mathbf{X}_t = \left[ f(\mathbf{X}_t, t) - g(t)^2 s_\theta(\mathbf{x}_t) \right] dt + g(t)d\mathbf{B}_t \tag{19}$$

As demonstrated by Song et al. (2021), diffusion models can be formulated as a special case of this framework under the VP-SDE formulation. For clarity in our analysis, we will use Equation 18 throughout.

The reverse SDE can be further controlled by incorporating a potential function $\log \Phi_t(\cdot)$ alongside the score function (Corso et al., 2024). This potential function introduces vector fields $(\nabla \log \Phi_t(\cdot))$ that interact with the score function $(\nabla \log p_t(\cdot))$ to provide enhanced control over the sampling process. The potential term can be selected based on specific requirements—for example, Corso et al. (2024) employed a kernel-based potential to increase the diversity of generated samples. Formally, this modification yields:

$$d\mathbf{X}_t = \left[ f(\mathbf{X}_t, t) - g(t)^2 \left( \nabla_{\mathbf{x}} \log p_t(\mathbf{x}) + \nabla_{\mathbf{x}} \log \Phi_t(\mathbf{x}) \right) \right] dt + g(t)d\mathbf{B}_t \tag{20}$$

Our proposed debiasing method can be analyzed within this extended reverse SDE framework.

## A.2  Analysis of SG-Exemplar

Consider a sequence $\{\mathbf{e}_t\}$ that converges to the conditional expectation, i.e., $\mathbf{e}_t \xrightarrow{t} \mathbb{E}[\mathbf{X}|\mathbf{Y} = \mathbf{a}]$. We define $r$-radius neighborhoods around each point in this sequence as $B_t \triangleq \{x \in \mathcal{X} \mid \|x - \mathbf{e}_t\| \leq r\}$.

For analytical purposes, we examine a variant of our method where guidance is performed in the data space rather than the **H**-space. This variant is formulated as:

$$\hat{\mathbf{x}}_{0|t} = \hat{\mathbf{x}}_{0|t} - \frac{\sqrt{\alpha_t}}{1 - \bar{\alpha}_t} \left( \hat{\mathbf{x}}_{0|t} - \mathbf{e}_t \right) \mathsf{ReLU} \left( 1 - \frac{r}{\|\hat{\mathbf{x}}_{0|t} - \mathbf{e}_t\|} \right) \tag{21}$$

The $\mathsf{ReLU}(\cdot)$ function ensures that the update is applied only when $\hat{\mathbf{x}}_{0|t}^{(i)}$ lies outside the neighborhood $B_t$. This update term can be interpreted as $\nabla_{\mathbf{x}} \log p_{0|t}(\mathbf{X}_0 \in B_t \mid \mathbf{X}_t)$, representing the direction necessary to ensure that the predicted denoised sample remains within the neighborhood $B_t$.

**Theorem A.1.** *For the guidance mechanism defined in Equation 21, the following bound holds:*

$$\| \mathbb{E}[\mathbf{X}_0] - \mathbb{E}[\mathbf{X}|\mathbf{Y} = \mathbf{a}]\| \leq r \quad \text{with probability 1} \tag{22}$$

*Proof.* We begin by applying Bayes' theorem to express the gradient of the log conditional probability:

$$\nabla_{\mathbf{x}_t} \log p_t(\mathbf{x}_t \mid \mathbf{X}_0 \in B_t) = \nabla_{\mathbf{x}_t} \log p_t(\mathbf{x}_t) + \nabla_{\mathbf{x}_t} \log p_{0|t}(\mathbf{X}_0 \in B_t \mid \mathbf{X}_t) \tag{23}$$

Using this result, the reverse SDE obtained via our guidance mechanism becomes:

$$dX_t = \left[f(X_t, t) - g(t)^2 \nabla_{x_t} \log p_t(x_t | X_0 \in B_t)\right] dt + g(t) dB_t \tag{24}$$

$$= \left[f(X_t, t) - g(t)^2 \left(\nabla_{x_t} \log p_t(x_t) + \nabla_{x_t} \log p_{0|t}(X_0 \in B_t \mid X_t)\right)\right] dt + g(t) dB_t \tag{25}$$

The term $\nabla_{x_t} \log p_{0|t}(X_0 \in B_t \mid X_t)$ can be interpreted as Doob's h-transform. Next, applying Tweedie's formula to the modified SDE yields:

$$\mathbb{E}[X_0 | x_t] = \frac{x_t + (1 - \bar{\alpha}_t)\left(\nabla_{x_t} \log p_t(x_t) + \nabla_{x_t} \log p_{0|t}(X_0 \in B_t \mid X_t)\right)}{\sqrt{\bar{\alpha}_t}} \tag{26}$$

$$= \frac{x_t + (1 - \bar{\alpha}_t)\nabla_{x_t} \log p_t(x_t)}{\sqrt{\bar{\alpha}_t}} + \frac{1 - \bar{\alpha}_t}{\sqrt{\bar{\alpha}_t}} \nabla_{x_t} \log p_{0|t}(X_0 \in B_t \mid X_t) \tag{27}$$

$$= \hat{x}_{0|t} - \frac{1 - \bar{\alpha}_t}{\sqrt{\bar{\alpha}_t}} \times \frac{\sqrt{\bar{\alpha}_t}}{1 - \bar{\alpha}_t} \left(\hat{x}_{0|t} - e_t\right) \mathsf{ReLU}\left(1 - \frac{r}{\|\hat{x}_{0|t} - e_t\|}\right) \tag{28}$$

$$= \hat{x}_{0|t} - \left(\hat{x}_{0|t} - e_t\right) \mathsf{ReLU}\left(1 - \frac{r}{\|\hat{x}_{0|t} - e_t\|}\right) \tag{29}$$

We now analyze two cases:

**Case 1:** $\|\hat{x}_{0|t} - e_t\| \leq r$

In this case, the $\mathsf{ReLU}(\cdot)$ term becomes zero, resulting in $\mathbb{E}[X_0 \mid x_t] = \hat{x}_{0|t}$. Since we've assumed $\|\hat{x}_{0|t} - e_t\| \leq r$, it follows that $\|\mathbb{E}[X_0 \mid x_t] - e_t\| \leq r$.

**Case 2:** $\|\hat{x}_{0|t} - e_t\| > r$

In this case:

$$\|\mathbb{E}[X_0 \mid x_t] - e_t\| = \left\|\hat{x}_{0|t} - \left(\hat{x}_{0|t} - e_t\right)\left(1 - \frac{r}{\|\hat{x}_{0|t} - e_t\|}\right) - e_t\right\| \tag{30}$$

$$= \left\|e_t + \frac{r(\hat{x}_{0|t} - e_t)}{\|\hat{x}_{0|t} - e_t\|} - e_t\right\| \tag{31}$$

$$= \left\|\frac{r(\hat{x}_{0|t} - e_t)}{\|\hat{x}_{0|t} - e_t\|}\right\| \tag{32}$$

$$= r \tag{33}$$

Therefore, for all time steps $t$, we have $\|\mathbb{E}[X_0 \mid x_t] - e_t\| \leq r$.

Given that $e_t \xrightarrow{t} \mathbb{E}[X \mid Y = a]$, for any $\epsilon > 0$, $\exists t'$ such that for all $t > t'$, we have $\|e_t - \mathbb{E}[X \mid Y = a]\| < \epsilon$.

By applying the triangle inequality, for all $t > t'$:

$$\|\mathbb{E}[X_0 \mid x_t] - \mathbb{E}[X \mid Y = a]\| \leq \|\mathbb{E}[X_0 \mid x_t] - e_t\| + \|e_t - \mathbb{E}[X \mid Y = a]\| \tag{34}$$

$$\leq r + \epsilon \tag{35}$$

Since $\epsilon$ is arbitrary, we can make it arbitrarily small. As $t \to \infty$, we have $\mathbb{E}[X_0 \mid x_t] \to \mathbb{E}[X_0]$ and $\|\mathbb{E}[X_0 \mid x_t] - \mathbb{E}[X \mid Y = a]\| \leq r$ with probability 1, hence, $\|\mathbb{E}[X_0] - \mathbb{E}[X \mid Y = a]\| \leq r$ with probability 1 which completes the proof. $\square$

The above theorem provides a powerful guarantee: the adjusted SDE will generate samples in the vicinity of $\mathbb{E}[X \mid Y = a]$. More specifically, it ensures that as $t$ increases, the expected value of the generated sample will remain within a distance $r$ of the true conditional expectation $\mathbb{E}[X \mid Y = a]$.

This result has significant implications for controlled generation. By formulating our guidance mechanism through the potential function that enforces proximity to the evolving sequence $\{\mathbf{e}_t\}$, we establish a theoretical bound on how far the generated samples can deviate from the desired target. The parameter $r$ effectively controls the trade-off between diversity and accuracy—smaller values of $r$ produce samples closer to the conditional expectation but with potentially less diversity, while larger values permit more variation in the generated outputs while still maintaining statistical fidelity to the conditioning information.

## B  Implementation Details

Here we provide details regarding implementation of the proposed method. Particularly, we expand on the details of text-based debiasing and exemplar-based debiasing for transparency. The code can be accessed through *codebase*

### B.1  Text-based Debiasing

As mentioned in the main text, we use CLIP for score-guidance. Moreover, we update $\mathbf{H}$-space vectors for this guidance as shown in Eq. 10. For Stable Diffusion, we use the decoded latent (usign decoder of Stable Diffusion) to calculate CLIP similarity with attribute texts. The attribute text used for debiasing for different attributes are provided in Table 7. There are four hyper-parameters used for debiasing - $M$ (number of times the update is applied at each step), $\gamma$ (guidance strength), $(t_{\text{start}}, t_{\text{end}})$ (time window in which the update is applied). These values are provided in Table 8.

For multi-class attributes, we tag samples in an iterative process. Consider an attribute with $n$ classes and reference distribution $\mathbf{p}_{\mathbf{ref}}^{\mathbf{a}} = [p_1, \ldots, p_n]$ for a batch of size $B$. First, we identify the top $p_1 \cdot B$ samples with the highest CLIP-similarity to $\mathbf{t}_1$ and tag them with $a_1$. From the remaining $(1 - p_1) \cdot B$ samples, we select the top $p_2 \cdot B$ samples with highest CLIP-similarity to $\mathbf{t}_2$ and tag them with $a_2$. This process continues until all samples are tagged according to the desired proportions. For multi-attribute debiasing, tagging follows a recursive approach. For example, with two attributes having reference distributions $\mathbf{p}_{\mathbf{ref}}^{\mathbf{a}_1} = [p_1, p_2]$ and $\mathbf{p}_{\mathbf{ref}}^{\mathbf{a}_2} = [p_3, p_4]$, we first tag samples for $\mathbf{a}_1$ in the ratio $p_1 \cdot B$ to $p_2 \cdot B$ (for batch size $B$). Then, we further tag the $p_1 \cdot B$ samples (already tagged as $\mathbf{a}_1$) for $\mathbf{a}_2$ in the ratio $p_1 \cdot p_3 \cdot B$ to $p_1 \cdot p_4 \cdot B$, achieving the desired distribution.

### B.2  Exemplar-based Debiasing

For exemplar-based debiasing, we take exemplar images from the dataset itself. For uncnditional P2 model, we take samples from CelebA-HQ[6]. For conditional Stable Diffusion, we explicitly generate exemplar images by passing prompts like 'image of a female firefighter'[7]. We use only eight exemplar images for all the experiments.

Further, the image space update to push the predicted denoised samples inside a $r$-ball radius of anchor points is given by:

$$\hat{\mathbf{x}}_{0|t}^{(i)} = \hat{\mathbf{x}}_{0|t}^{(i)} - \left(\hat{\mathbf{x}}_{0|t}^{(i)} - \bar{\mathbf{e}}_{0|t}^{(j)}\right)\left(1 - \frac{r}{\|\hat{\mathbf{x}}_{0|t}^{(i)} - \bar{\mathbf{e}}_{0|t}^{(j)}\|}\right) \tag{36}$$

however, instead of updating $\hat{\mathbf{x}}_{0|t}^{(i)}$, we update the associate $\mathbf{H}$-space vectors:

$$\mathbf{h}^{(i)} = \mathbf{h}^{(i)} - \gamma\nabla_{\mathbf{h}^{(i)}}\left(\hat{\mathbf{x}}_{0|t}^{(i)} - \bar{\mathbf{e}}_{0|t}^{(j)}\right)\left(1 - \frac{r}{\|\hat{\mathbf{x}}_{0|t}^{(i)} - \bar{\mathbf{e}}_{0|t}^{(j)}\|}\right) \tag{37}$$

where $\gamma$ is the guidance strength. We again apply this update $M$ times at each time step in the time window $\mathcal{T}$. These hyperparameters for different attributes are provided in Table 9. The tagging process is carried out

---

[6]we use CLIP similarity with images to randomly collect exemplar images

[7]we do this to make sure that exemplar images are similar to the samples generated by these models

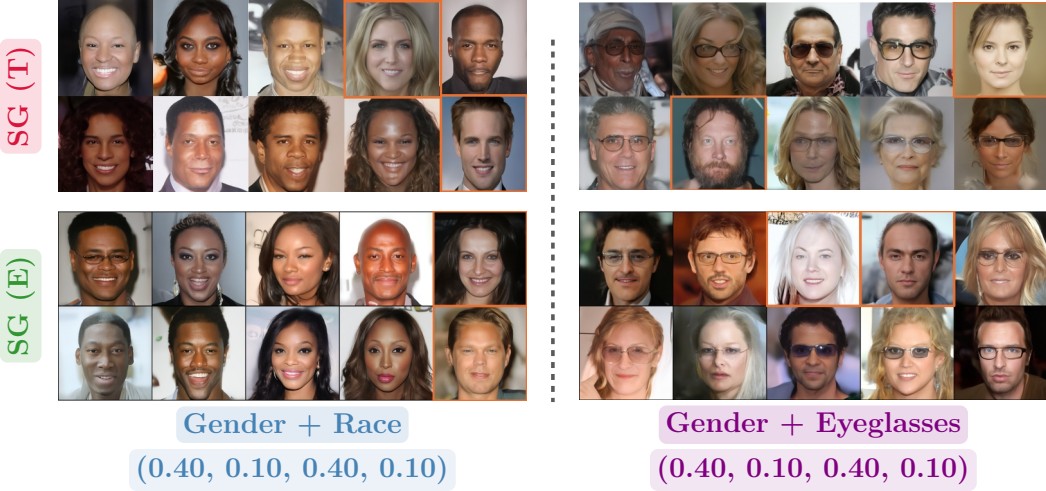

Figure 6: Visual results for skewed reference distributions using SG for multiple attributes.

in similar manner as Text-based debiasing, except instead of CLIP similarity we consider $\ell_2$-distance from anchor points.

Table 7: Attribute texts used for text-based score guidance for debiasing

| Attribute | Attribute Text |
|---|---|
| Race (2 classes) | 'image of a black person', 'image of a white person' |
| Race (4 classes) | 'image of a black person', 'image of a white person', 'image of an asian person', 'image of a brown person' |
| Gender | 'a male person', 'a female person' |
| Eyeglasses | 'image of a person with glasses', 'image of a person without glasses' |
| Age (3 classes) | 'a very young child', 'a middle aged adult', 'a very old person' |

Table 8: Hyper-parameter values for SG (T)

| Attributes | M | $\gamma$ | $t_{\text{end}}$ | $t_{\text{start}}$ |
|---|---|---|---|---|
| Race (2 classes) | 3 | $1.0 \times 10^6$ | 18 | 46 |
| Race(4 classes) | 7 | $1.0 \times 10^6$ | 14 | 50 |
| Gender | 5 | $1.0 \times 10^6$ | 36 | 50 |
| Eyeglasses | 7 | $1.1 \times 10^6$ | 22 | 50 |
| Age (3 classes) | 2 | $5.0 \times 10^5$ | 12 | 44 |

Table 9: Hyper-parameter values for SG (E)

| Attributes | M | $\gamma$ | $t_{\text{end}}$ | $t_{\text{start}}$ |
|---|---|---|---|---|
| Race (2 classes) | 1 | 1.0 | 16 | 50 |
| Race (4 classes) | 1 | 1.5 | 2 | 50 |
| Gender | 1 | 1.0 | 2 | 50 |
| Eyeglasses | 1 | 0.8 | 2 | 50 |
| Age (3 classes) | 1 | 1.0 | 2 | 50 |

Table 10: Results on Imbalanced Generation with Multiple attributes

| Method | Gender + Eyeglasses (0.40, 0.10, 0.40, 0.10) | | Gender + Race (0.40, 0.10, 0.40, 0.10) | |
|---|---|---|---|---|
| | FD (↓) | FID (↓) | FD (↓) | FID (↓) |
| Random Sampling | 1.100 | 49.45 | 1.444 | 49.45 |
| Sample Guidance (Parihar et al., 2024) | 0.472 | 48.66 | 0.756 | 62.48 |
| Distribution Guidance (Parihar et al., 2024) | 0.380 | 47.68 | 0.464 | 45.51 |
| SG – Text | 0.258 | 48.33 | 0.148 | 46.61 |
| SG – Exemplar | 0.147 | 42.87 | 0.160 | 44.17 |

### B.3 Evaluation Classifier

As mentioned in Section 4, we use Fairness Discrepancy to evaluate our method. This requires a high accuracy classifier $\mathcal{C}_{\mathbf{a}}$. We use a ResNet-18 based classifier for this. To train the classifier, we use the training dataset itself. E.g., to train a classifier on 'gender', we use the provided labels in CelebA-HQ to train the classifier. For attributes whose labels are not available (e.g., 'race'), we use CLIP similarity to create such dataset and train the classifier on that dataset.

## C Other Results and Visualization

### C.1 Robustness of Sample Tagging

In this section, we discuss and elaborate on the robustness of our sample tagging mechanism, which is a critical component of both text-based (SG (T)) and exemplar-based (SG (E)) guidance. We analyze the potential impact of biases in CLIP for SG (T) and the sensitivity to exemplar selection for SG (E).

Table 11: Results for balanced generation on multiple attributes

| Method | Gender + Race | | Eyeglasses + Race | | Gender + Eyeglasses | | Gender + Eyeglasses + Race | |
|---|---|---|---|---|---|---|---|---|
| | FD (↓) | FID (↓) | FD (↓) | FID (↓) | FD (↓) | FID (↓) | FD (↓) | FID (↓) |
| Random Sampling | 0.256 | 60.68 | 0.292 | 89.14 | 0.214 | 70.97 | 0.768 | 49.45 |
| H-Sample Guidance (Parihar et al., 2024) | 0.241 | 59.78 | 0.135 | 67.87 | 0.079 | 52.03 | 0.496 | 47.83 |
| H-Distribution Guidance (Parihar et al., 2024) | 0.075 | 49.91 | 0.101 | 57.46 | 0.057 | 52.03 | 0.408 | 43.94 |
| SG - Text (ours) | 0.173 | 50.81 | 0.132 | 54.90 | 0.062 | 49.33 | 0.132 | 41.72 |
| SG - Exemplar (ours) | 0.028 | 46.83 | 0.125 | 51.30 | 0.051 | 45.42 | 0.187 | 43.40 |

Figure 7: Visual results for balanced generation on multi-class attributes from Unconditional Diffusion using SG(T) and SG(E).

### C.1.1   Robustness of SG - Text to CLIP Biases

A valid concern regarding text-based guidance is the reliability of sample tagging using CLIP, given that CLIP text encoders are known to exhibit social biases (Tanjim et al., 2024; Hirota et al., 2024; Chuang et al., 2023; Berg et al., 2022; Wang et al., 2022). For instance, the text embedding for 'firefighter' may be closer to 'male' than 'female' in the embedding space.

However, our approach fundamentally differs from methods that rely on text-to-text similarity. We instead leverage CLIP's demonstrated and powerful zero-shot classification capabilities (Qian & Hu, 2024; Sammani & Deligiannis, 2024; Radford et al., 2021b; Sharma et al., 2025). Specifically, we tag samples by computing the similarity between the *image embedding* of $\hat{\mathbf{x}}_{0|t}$ and the *text embeddings* of the attribute classes (e.g., 'a photo of a male' vs. 'a photo of a female'). This text-to-image classification approach has proven highly reliable across numerous studies and avoids the direct conditioning biases that emerge from using text-only embeddings.

To validate this distinction, we conducted an experiment comparing the performance of our SG (T) method using the standard, off-the-shelf vanilla CLIP model versus a 'debiased' CLIP variant from prior work of Bansal et al. (2022). The results, presented in Table 12, demonstrate the robustness of our approach.

Interestingly, while the debiased CLIP showed a marginal improvement for the 'eyeglasses' attribute, it performed slightly worse for the 'gender' and 'race' attributes. More significantly, it consistently produced

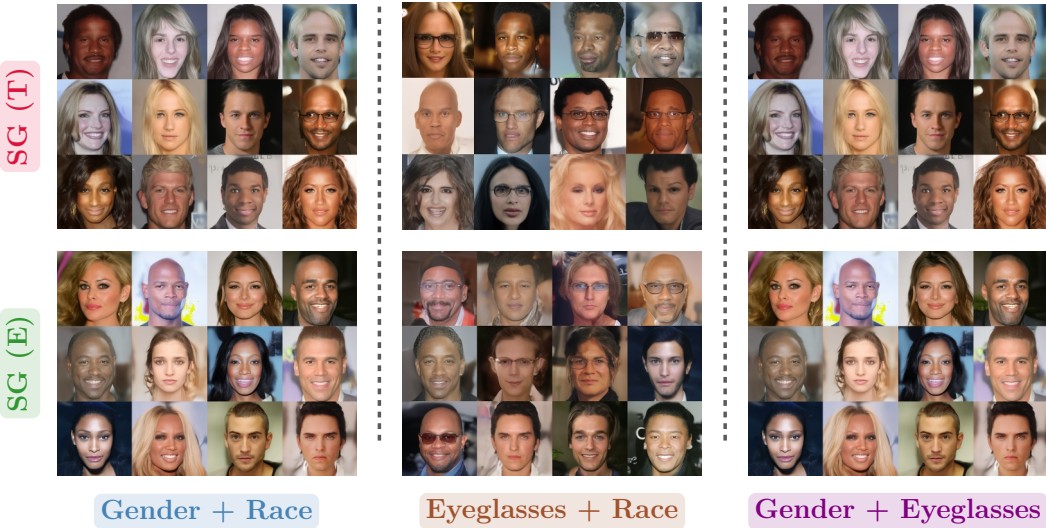

Figure 8: Visual results for balanced generation on multiple attributes for Unconditional Diffusion using SG(T) and SG(E).

lower image quality (higher FID scores). This is because many CLIP debiasing techniques involve fine-tuning the text embeddings without ensuring their continued alignment with the visual embeddings. This can disrupt the precise visual-textual correspondence that our zero-shot classification mechanism relies upon, ultimately degrading performance.

### C.1.2 Robustness of SG - Exemplar

For exemplar-based guidance, we analyze robustness from two perspectives: the **quantity** and **quality** of the exemplar samples used to compute the anchor points $\bar{\mathbf{e}}_{0|t}$.

**Quantity of Exemplar Samples**: As stated in the main text (Eq. 13), the anchor points serve as estimates for conditional expectations. By the Law of Large Numbers, these estimates converge to the true conditional expectation as the number of exemplar samples $k$ increases. While theoretical convergence requires $k \to \infty$, practical constraints necessitate finding an optimal balance between computational feasibility and guidance accuracy.

We performed a systematic ablation study on gender debiasing to demonstrate this relationship empirically. The results are shown in Table 13.

The results clearly demonstrate that both fairness (FD) and image quality (FID) improve consistently as the number of exemplar samples increases. This directly validates our theoretical framework. Importantly, we observe diminishing returns beyond $k = 8$, suggesting that our chosen default provides an effective balance between performance and efficiency. The results in Table 13 represent averages across three independent runs using randomly selected exemplar samples, demonstrating the robustness of our approach to sampling variance.

Further, to analyze if SG - Exemplar suffers through mode-collapse due to finite number of exemplar samples, we also calculate Intra-cluster LPIPS distance (Zhang et al., 2018) to analyze the diversity of generated samples / intra-class diversity. Particularly, following Ojha et al. (2021); Mondal et al. (2023), we assign 1000 generated images to one of the $k$ possible clusters (for $k$ exemplar samples) based on the lowest LPIPS distance (Zhang et al., 2018). Next, we compute the average pair-wise LPIPS metric among the members of the same cluster. Finally, we take the average over the $k$ clusters. A method will have a zero score if it generates same image every time. We present this metric with different values of $k$ in Table 14. On expected lines, with just a single exemplar sample ($k = 1$), there is indeed a mode-collapse since the generation is biased towards a single sample. However, with increasing $k$, LPIPS metric improves consistently because of the

Table 12: Comparison of SG (T) with vanilla CLIP and a debiased CLIP variant (Bansal et al., 2022).

| Method | Gender | | Race | | Eyeglasses | |
|---|---|---|---|---|---|---|
| | FD ($\downarrow$) | FID ($\downarrow$) | FD ($\downarrow$) | FID ($\downarrow$) | FD ($\downarrow$) | FID ($\downarrow$) |
| **SG - Text (w/ Vanilla CLIP)** | **0.022** | **35.24** | **0.093** | **43.08** | 0.116 | **55.24** |
| **SG - Text (w/ Debiased CLIP)** | 0.025 | 50.66 | 0.102 | 45.37 | **0.110** | 58.41 |

Table 13: Ablation on the number of exemplar samples ($k$) for gender debiasing.

| # Exemplar Samples ($k$) | FD ($\downarrow$) | FID ($\downarrow$) |
|---|---|---|
| **1** | 0.241 | 80.66 |
| **8** (default) | **0.00146** | **34.61** |
| **20** | 0.00092 | 32.53 |
| **50** | 0.00073 | 30.19 |

Table 14: Intra-cluster pairwise LPIPS distance for different value of $k$.

| # Exemplar Samples ($k$) | LPIPS distance ($\uparrow$) |
|---|---|
| **1** | 0.15 |
| **8** (default) | **0.64** |
| **20** | 0.71 |
| **50** | 0.74 |

Table 15: Comparison of Intra-cluster pairwise LPIPS distance for different methods.

| Method | LPIPS distance ($\uparrow$) |
|---|---|
| **Random** | 0.61 |
| **Balancing Act** (Parihar et al., 2024) | 0.57 |
| **SG - Text** | **0.66** |
| **SG - Exemplar** | 0.64 |

Table 16: Generation time (s) and peak GPU memory usage (GB) per-image with batch size 16 on NVIDIA RTX A6000.

| Method | Time (s) | Peak GPU Memory Usage (GB) |
|---|---|---|
| **Random** | 1.736 | 3.21 |
| **Balancing Act** (Parihar et al., 2024) | 9.427 | 7.44 |
| **SG - Text** | 4.211 | 4.10 |
| **SG - Exemplar** | 12.941 | 10.27 |

better estimate of conditional expectation as mentioned above. We compare the Intra-cluster LPIPS distance for different methods with $k = 8$ in Table 15. It can be seen that the LPIPS distance of SG-Exemplar closely outperforms the vanilla generation. We attribute this to two things: (i) explicit score guidance enabling exploration of additional modes beyond the original model distribution, and (ii) generation steps outside the guidance window, $\mathcal{T}$ encouraging diversity. Further, SG-Text achieves the highest diversity (0.66) as it avoids dependence on specific exemplars or classifiers.

**Quality of Exemplar Samples**: A second concern is the sensitivity of the anchor points to the *quality* or representativeness of the chosen exemplars. In all our experiments, we consistently employ *randomly selected* exemplar samples from the target attribute class. The fact that our method demonstrates significant improvements across all metrics (as shown in the main paper) using this random selection strategy actually strengthens our approach. It shows that SG - Exemplar does not require carefully curated or cherry-picked examples to achieve superior performance.

While deliberately choosing non-representative or poor-quality exemplars could certainly degrade performance, this would constitute an intentional misuse of the method rather than an inherent limitation. For applications requiring even higher quality assurance, our exemplar-based framework is fully compatible with quality enhancement techniques, such as using a reward model or RLHF to filter or select the 'best' exemplars. We leave this as an interesting direction for future work.

## C.2 Computational Requirements and Latency analysis

Since our method comprises of a guidance component, we provide comprehensive metrics to analyze the computational overhead incurred by Score Guidance. Specifically, we look at the time taken for generation (per-image) and the peak GPU memory usage. These metrics are provided in Table 16. SG-Text incurs a $7.5\times$ latency increase and $3.2\times$ memory increase compared to vanilla generation, primarily due to $M$ gradient steps and backpropagation through both UNet and CLIP encoders for computing gradients. We also note that, computational cost is comparable to Balancing Act (Parihar et al., 2024), while achieving superior performance across other metrics. Further, SG-Exemplar offers a more efficient alternative ($2.4\times$ latency, $1.3\times$ memory) while maintaining strong debiasing performance, making it suitable for resource-constrained deployments.

## C.3 Results on Skewed Generation for Multiple Attributes

While our main text presented results for single-attribute imbalanced generation, here we extend our analysis to scenarios involving multiple attributes simultaneously. For instance, consider the joint distribution of 'gender' and 'race' attributes with a skewed distribution of $(0.40, 0.10, 0.40, 0.10)$, corresponding to the

following demographic proportions: 40% Black male, 10% White male, 40% Black female, and 10% White female. Visual examples of such multi-attribute imbalanced generation are presented in Fig. 6.

The quantitative evaluation of our approach is detailed in Table 10. Our findings align with the single-attribute results, demonstrating that our method achieves superior performance in both Fréchet Distance (FD) and Fréchet Inception Distance (FID) across all test cases. The improvement in FD is particularly noteworthy – for the gender + eyeglasses combination, our method reduces FD to less than half compared to the second-best approach. Similar improvements are observed for the gender + race combination. These results further validate the effectiveness of our proposed method in handling multi-attribute imbalanced generation.

### C.4 Results on Multi-class and Multi-attribute Balanced Generation

We presented the quantitative results on multi-class attributes in Table 5 of main text. We provide qualitative result of the same in Fig. 7, where we observe that all the classes are appropriately generated in equal proportion.

Further, we extend the result on multi-attribute balancing to three attributes in Table 11. Our observation is consistent with two attribute balancing, where SG (T) provides the best FD and FID whereas SG (E) performs second best. Further, it is also observed that the improvement in FD for SG is significant as compared to **H**-distribution guidance and other baselines. We provide qualitative results for our method in Fig. 8.

## D Visualization of Debiasing in Stable Diffusion

We show the debiasing results for Stable Diffusion through visualizations , for gender-balanced generations across images of classes *Taxi Driver, CEO, Artist, Doctor, Firefighter* in Figure 9, and for (race,gender)-balanced generations across classes *Teacher, Nurse, Artist, Taxi Driver* in Figure 10. We tested both Text-based (SG - Text) and Exemplar-based (SG - Exemplar) score-guidance methods. Our results show that different methods tag different samples for guidance, likely because they use different tagging approaches. Specifically, SG (Exemplar), which uses $\ell_2$ norm-based tagging, often picks different samples compared to SG (Text), which uses CLIP-based tagging. We also found that SG (Text) often makes big changes to the original Stable Diffusion-generated samples, while SG (Exemplar) makes smaller, more subtle changes. This difference in how the methods modify images might explain why SG (Exemplar) performs better than SG (Text).

## E Visualization of Debiasing in SD2.0 and SDXL

We provide the debiasing results for SDv2.0 Stability AI (2022) and SDXL Podell et al. (2023) for the profession of 'Doctor' and 'Firefighter' respectively. The quantitative metrics are provided in Table 17 and 18 respectively. It can be observed that FD is highly skewed in random sampling, indicating biased generation. However, after applying the proposed Score Guidance, we observe significant reduction in FD while maintaining the generation quality as indicated through FID.

We also visualize the debiasing results for SDv2.0 and SDXL. Figures 11 and 12 present the results for the two models. It can be seen that the random generations from both models exhibit a strong bias towards the 'male' attribute. In contrast, after applying the Score Guidance, the generated images reflect an equal proportion of 'male' and 'female' attributes while preserving image quality, confirming the effectiveness of our approach.

## F Ablation Results

In this section, we present the ablation results to study the effect of different hyper-parameters on SG-based debiasing. Specifically, we study the impact of three hyper-parameters: $M$ (number of updates per step), $\gamma$ (learning rate of the updates) and $\mathcal{T}$ (time window in which the updates are performed). We consider gender balancing to study these effects for both SG (T) and SG (E).

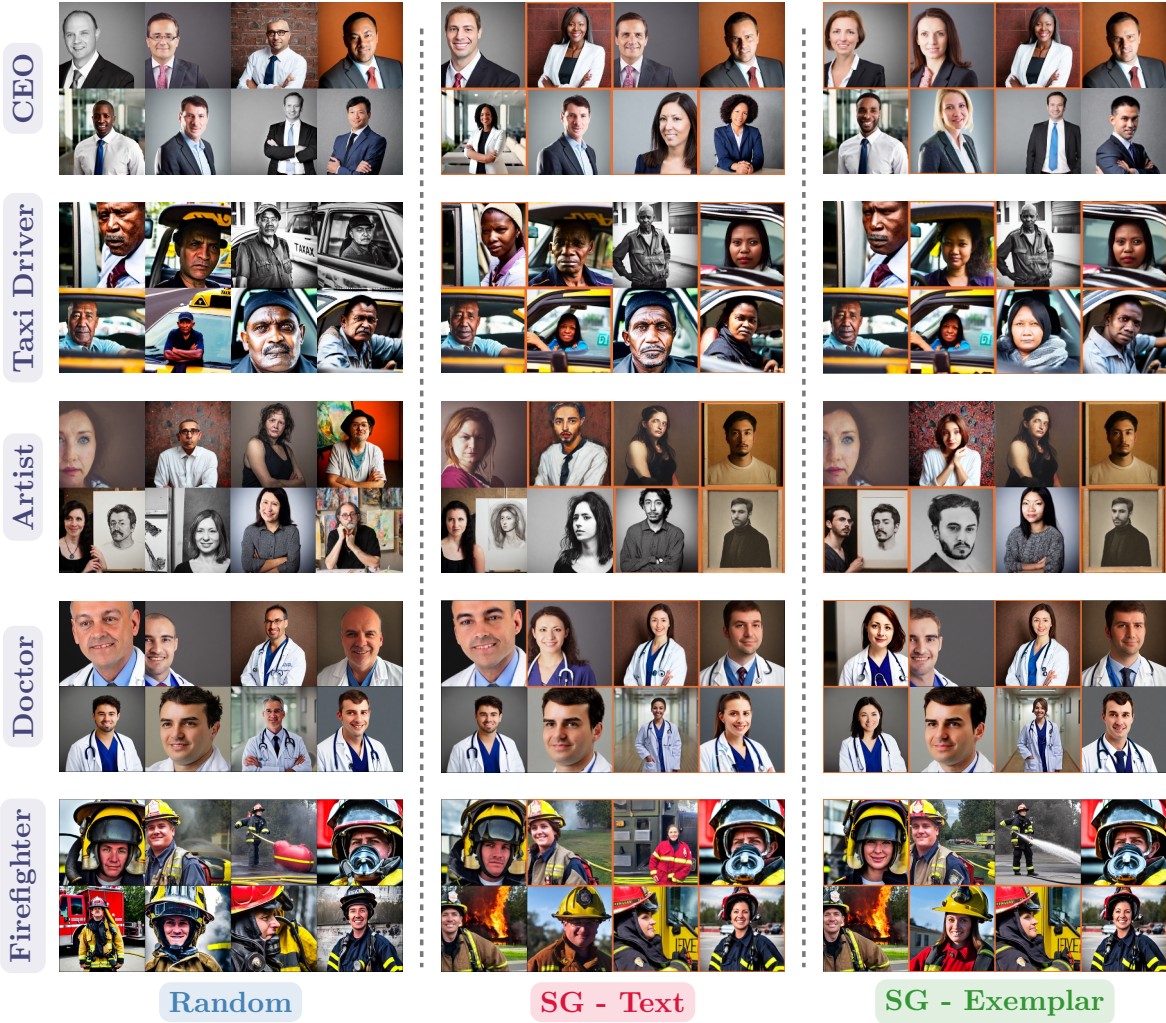

Figure 9: Visualizations of 'gender-balanced' samples for different profession from Stable Diffusion using SG (T) and SG (E).

Table 17: Results for debiased generation for SDv2.0

| Method | Doctor | | Firefighter | |
|---|---|---|---|---|
| | FD ($\downarrow$) | FID ($\downarrow$) | FD ($\downarrow$) | FID ($\downarrow$) |
| Random Sampling | 0.331 | 67.22 | 0.484 | 70.19 |
| SG - Text (ours) | 0.094 | 63.99 | 0.107 | 64.41 |
| SG - Exemplar (ours) | 0.012 | 58.00 | 0.033 | 61.56 |

Table 18: Results for debiased generation for SDXL

| Method | Doctor | | Firefighter | |
|---|---|---|---|---|
| | FD ($\downarrow$) | FID ($\downarrow$) | FD ($\downarrow$) | FID ($\downarrow$) |
| Random Sampling | 0.277 | 22.57 | 0.286 | 24.18 |
| SG - Text (ours) | 0.111 | 20.05 | 0.085 | 18.73 |
| SG - Exemplar (ours) | 0.005 | 18.74 | 0.013 | 18.61 |

We present these results in Fig. 13 and Fig. 14. Specifically, we plot the FD v/s FID graph for these hyper-parameters. An ideal debiasing method should provide FD=0 and FID=0. For $M$, we observe that lower values lead to better FID but suffer in terms of FD. Conversely, higher $M$ leads to better FD but higher FID. This can be explained as follows: a higher $M$ - more updates per step - would align the score more closely with the desired classes, leading to better balancing. Meanwhile, a lower $M$ - fewer updates per step - would not force such alignment. This explains the better FD for higher $M$. However, over-alignment also affects generation quality as CLIP (or exemplar images) begins to dominate the generation process rather than following the natural path of the diffusion model. Hence, a moderate value of $M$ is optimal, as shown in the figures.

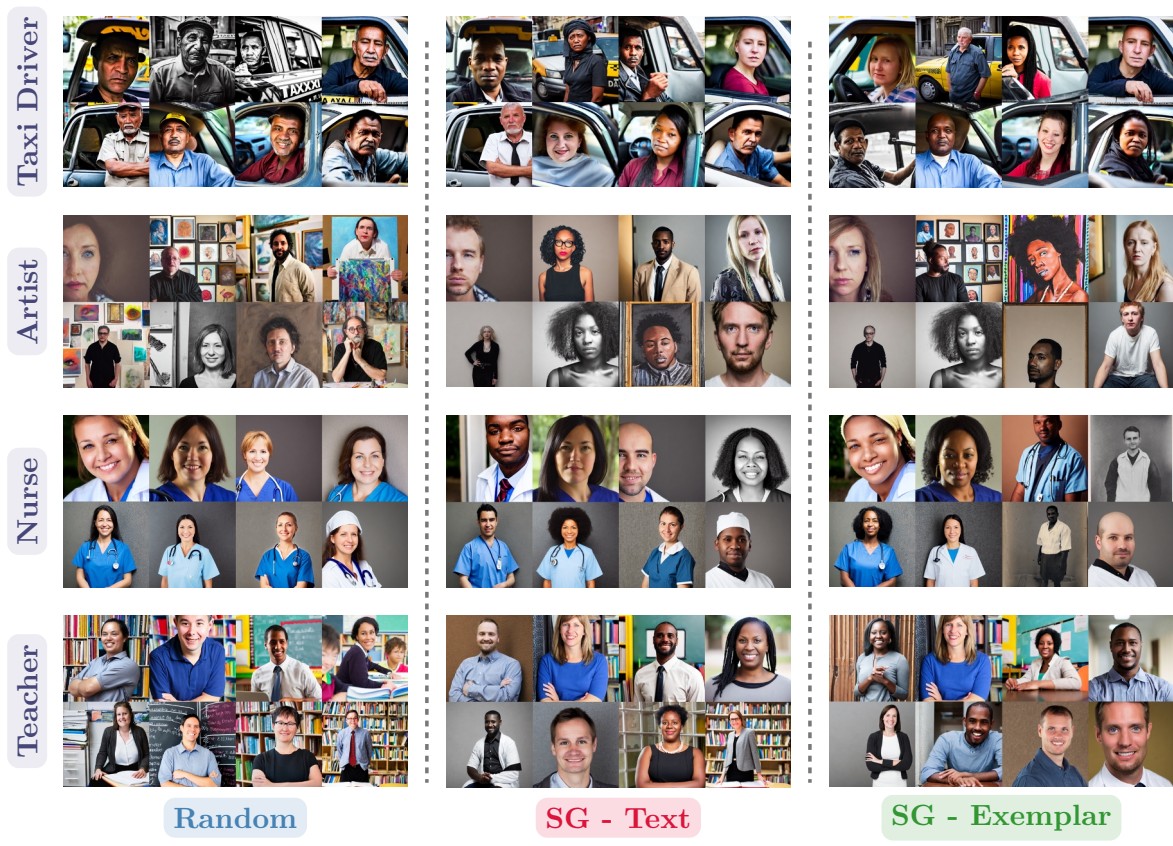

Figure 10: Visualizations of multi attribute ('race' and 'gender') debiasing for different profession from Stable Diffusion using SG (T) and SG (E).

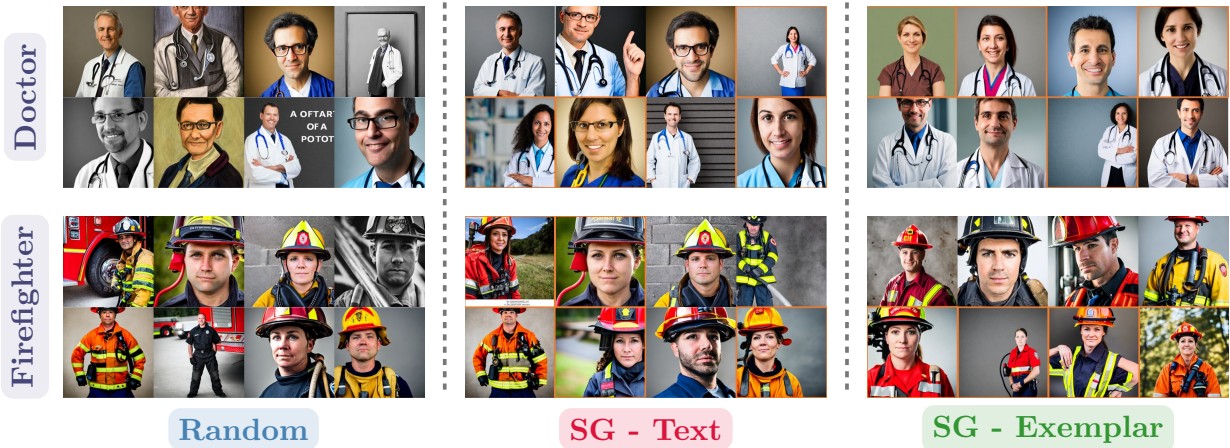

Figure 11: Visualizations of debiasing for 'doctor' and 'firefighter' from Stable Diffusion v2.0 using SG (T) and SG (E).

Next, we observe that $\gamma$ has a positive correlation with both FD and FID - both increase as $\gamma$ increases. Since $\gamma$ functions as the learning rate in the updates, it effectively controls the sensitivity toward the gradient signal provided by CLIP or exemplar images. A higher learning rate can lead to improper scaling of gradients, resulting in suboptimal solutions and consequently worse FID and FD.

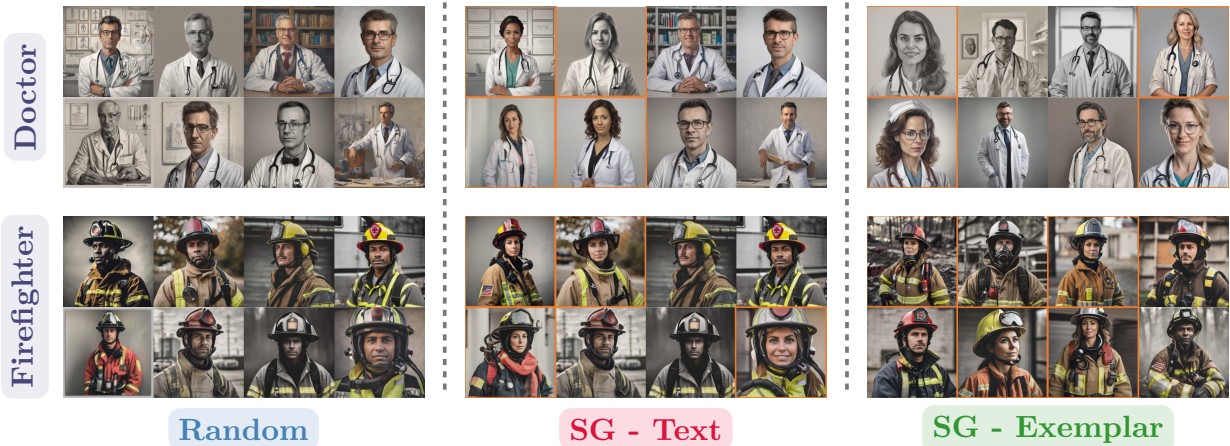

Figure 12: Visualizations of debiasing for 'doctor' and 'firefighter' from Stable Diffusion XL using SG (T) and SG (E).

Lastly, we examine the effect of the time window in the third plot of both figures. We find that longer time windows lead to better FD but higher FID, while shorter time windows produce better FID but higher FD. This occurs because longer time windows allow more updates to the score, creating stronger alignment with the desired classes. However, such over-alignment can deteriorate quality. Conversely, shorter time windows mean fewer updates, which preserves quality but compromises score alignment. Therefore, one can choose to trade off between these metrics depending on the desired outcome.

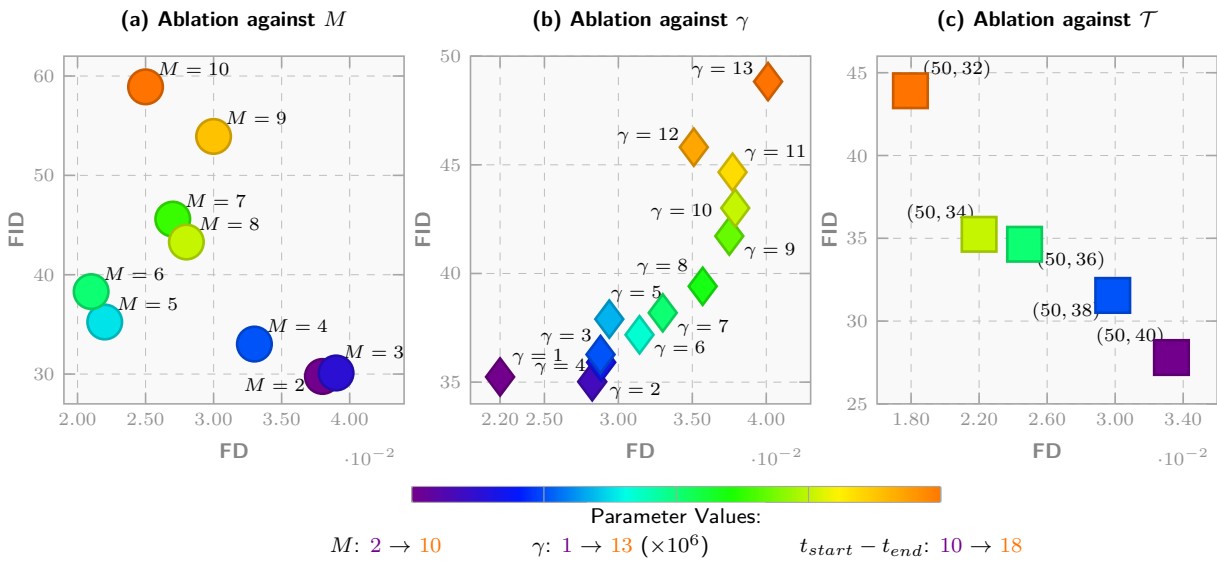

Figure 13: Ablation studies on model parameters $M$, $\gamma$, and $\mathcal{T}$ for MMSG-Text. Each plot shows FD vs FID performance with different parameter values.

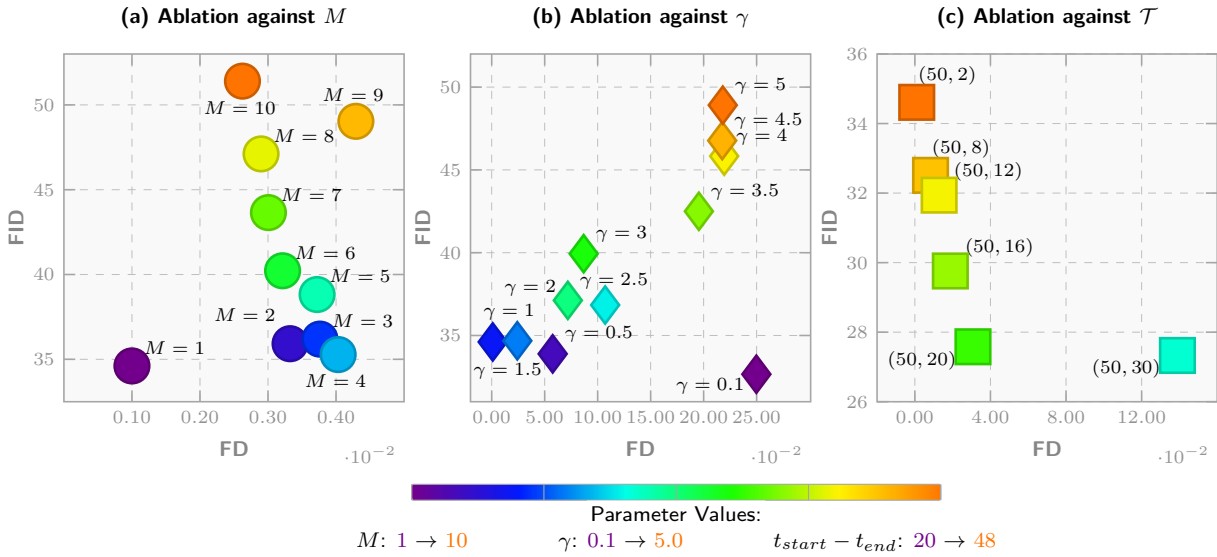

Figure 14: Ablation studies on model parameters $M$, $\gamma$, and $\mathcal{T}$ for MMSG-Exemplar. Each plot shows FD vs FID performance with different parameter values.

