# OpenReview forum: "Debiasing Diffusion Models via Score Guidance"
_TMLR — Accepted by TMLR_

### Review · Reviewer_TCYy · 2025-12-29

**Summary Of Contributions:**

The authors note the problem of distributional biases in diffusion based image generation models. They decompose the score-matching formulation into a weighted combination of different conditional distributions and use this to propose two steering methodologies that should move the generations towards some ideal (unbiased) distribution. One of them steers the generations of a diffusion model using a set of exemplar images for a given class and the other steers them based on textual descriptions of a class.

## Steering setup

1. __Text-based__: They use CLIP to embed the textual description (of a class) and the predicted clean images. The cosine distance between these embeddings is used to guide the generation process. They calculate the update in the bottleneck dimension of the U-Net.

2. __Exemplar-based__: They use DDIM-inversion on every image in the exemplar of every class to get latents and noise-predictions at every time step. They use this to predict the clean sample for every time step. The predicted clean samples are aggregated (by averaging) across all samples in a class to get the “anchor point” for a class at each time step — which is used to bias the generation process.


## Evaluation

To evaluate fairness, they use the FD (Fairness Discrepency) metric, that uses a trained image classifier to get softmax probabilities over the different classes, which are in-turn compared to the uniform vector to get a bias score.

They use the Frechet Inception Distance (FID) to evaluate the quality and diversity of generated samples.

# Strengths & Weaknesses

## Strengths

1. The problem is well-motivated
2. Mitigating distributional bias through steering makes sense
3. Steering methods seem sound and seem to work empirically

## Weaknesses

1. Equation 9 seems to be incomplete, specifically the $\nabla_{h^{(i)}}$ has not been introduced and it is unclear how the term is calculated.

2. The authors make the implicit assumption that the class conditioned distribution is uni-modal. This is assumption must be clearly stated. This is implied from how they obtain the "anchor points" (by averaging).

3. Results have been shown on only one model.

**Audience:**

Yes

**Audience Explanation:**

This paper will be interesting to bias researchers. Additionally, it will be interesting to anyone interested in controlling the generations from a diffusion model.

**Broader Impact Concerns:**

No concerns

**Claims And Evidence:**

Yes

**Claims Explanation:**

The burdens incurred by the authors are as follows:

1. The distribution of the generated samples needs to be shown to be reasonably unbiased.
2. The generated samples need to be high quality and diverse.

The burdens seem to be satisfied according to the experiment results.

**Requested Changes:**

1. Please introduce and clearly define $\nabla_{h^{(i)}}$ in equation 9  [IMPORTANT]
2. Please reproduce results on another model. [OPTIONAL]

---

> ### Author Response · Authors · 2026-01-25
>
> Thanks a lot for your time and thoughtful comments. These comments have helped us in improving our presentation. Please find below our response to your comments.
>
>
> #### 1. [Definition $\nabla_{h^{(i)}}$]
> We apologize for this oversight. $h^{(i)}$ denotes the $\textbf{H}$-space representation of the $i$-th sample in the batch. Specifically, $\hat{\mathbf{x}}^{(i)}\_{0|t}$ is related to $\epsilon\_\theta(\mathbf{x}^{(i)}\_t,t)$ via Eq. (5). Following [1], considering the UNet-type architecture, we can write the denoiser network as $\epsilon\_\theta(\mathbf{x}^{(i)}\_t,t) = D\_\theta \circ E\_\theta(\mathbf{x}^{(i)}\_t,t)$ where $D\_\theta(\cdot)$ and $E\_\theta(\cdot)$ are the decoder and encoder part of the network. The **H**-space representation can be denoted as: $h^{(i)} = E\_\theta(\mathbf{x}^{(i)}\_t,t)$. As shown in [1], **H**-space is more semantically meaningful and easier to drive the generation in diffusion models, we leverage this property for our purpose. $\nabla\_{h^{(i)}}$ represents the gradient with respect to this representation.
>
> #### 2. [Results on more Models]
> Thanks for this suggestion. We have shown the results for two models - an unconditional P2 model and a conditional T2I SDv1.5 model. As per your suggestion, we evaluate our method on two additional models - [Stable Diffusion v2.0 (SDv2.0)](https://huggingface.co/docs/diffusers/api/pipelines/stable_diffusion/stable_diffusion_2) and high-resolution supported [Stable Diffusion XL (SDXL)](https://huggingface.co/docs/diffusers/using-diffusers/sdxl). We provide the quantitative results below:
>
>
> **SDv2.0**:
> |                              | Doctor    |           | Firefighter      |    |
> |------------------------------|-----------|-----------|-----------|-----------|
> |                              | FD        | FID       | FD        | FID       |
> | Random                       | 0.331     | 67.220    | 0.484     | 70.192    |
> | SG - Text                    | 0.094     | 63.991    | 0.107     | 64.414    |
> | SG - Exemplar                | 0.012     | 58.001    | 0.033     | 61.569    |
>
>
> **SDXL**:
> |                              | Doctor    |           | Firefighter      |    |
> |------------------------------|-----------|-----------|-----------|-----------|
> |                              | FD        | FID       | FD        | FID       |
> | Random                       | 0.277     | 22.572    | 0.286     | 24.187    |
> | SG - Text                    | 0.111     | 20.058    | 0.085     | 18.733    |
> | SG - Exemplar                | 0.005     | 18.742    | 0.013     | 18.611    |
>
> From these results, we again observe similar trend as those seen with other diffusion models. The models inherently show a lot of bias as seen by the FD scores, our score guidance significantly reduces this bias while maintaining the image quality as measured by FID in both SDv2.0 and SDXL. Further we also provide qualitative results for both models: \
> SDv2.0 - [https://anonymous.4open.science/r/Debiased_gen-7024/rebuttal/sdv2.png](https://anonymous.4open.science/r/Debiased_gen-7024/rebuttal/sdv2.png) \
> SDXL - [https://anonymous.4open.science/r/Debiased_gen-7024/rebuttal/sdxl.png](https://anonymous.4open.science/r/Debiased_gen-7024/rebuttal/sdxl.png) \
> Here we observe that score guidance effectively mitigates biases inherent in default models. Further, with both models we see that the aesthetics are maintained without any artifact.
>
>
> #### 3. [Implicit assumption for true class-conditional score]
> Thanks for pointing this out. We indeed implicitly assume that we have access to a good and un-biased estimate of true class-conditional score function. We will mention this explicitly in the revised version.
>
> We will add all these results in the final draft.
>
>
> [1] Kwon, Mingi, Jaeseok Jeong, and Youngjung Uh. "Diffusion Models Already Have A Semantic Latent Space." The Eleventh International Conference on Learning Representations.

---

### Review · Reviewer_Kg9R · 2025-12-29

**Summary Of Contributions:**

This paper addresses the issue of inherent biases in Diffusion Models (DMs). The authors state that these biases creep in because of either biases in the training data, or in the labelling processes. They also theoretically show that the biases can be seen as originating because of uneven distribution of the dataset distribution which then influence the weighted convex combination of the global score function of the data. To mitigate this, the authors propose Score Guidance (SG), a training-free, inference-time framework. This method pushes generated samples towards a distribution which matches a target distribution specified by users. The framework operates via two modalities.
1. SG-Text: Uses CLIP similarity to "tag" samples based on textual descriptions and guides them using the gradient of CLIP similarity.
2. SG-Exemplar: Uses a small set of exemplars to create "anchor points" using DDIM inversion. Samples are guided to stay within a specific radius of these anchors to ensure attribute alignment.
Strengths:
1) The main advantage of this method is that it does not require any training and can be applied to any off-the-shelf diffusion UNet style model.
2) It works with both text (using CLIP embeddings) and image exemplars providing more flexibility in defining the target distribution.
3) The method is evaluated on unconditional models (P2 on CelebA-HQ) and conditional models (Stable Diffusion v1.5) and shows state-of-the-art performance compared to existing methods which reduce bias in DMs.
Weaknesses:
Please refer to the requested changes section

**Audience:**

Yes

**Audience Explanation:**

This work is useful for researchers in the fields of generative models, ethics and fairness research, and multi-modality research.

**Claims And Evidence:**

Yes

**Claims Explanation:**

The authors provide comprehensive evaluation results across unconditional generation, covering different ways such as balanced binary attributes, unbalanced binary attributes, multi-class attributes, and also conditional text-to-image generation. They use the FD and FID metrics which are the standard ones used by prior work as well. There are also sufficient qualitative examples provided for various types of attributes which show improvements vs baselines.They also include clear ablations for different hyper-parameter choices in the appendix.

**Requested Changes:**

The paper is well written overall and has clearly mentioned the theoretical aspects, experimental results and ablation sections. The following might help strengthen the work:

1) Sensitivity analysis and Intra-class diversity: For the SG-Exemplar method, the authors currently use k=8 exemplars from the dataset at random. It is important to evaluate robustness under adverse conditions, such as when exemplars are outliers or exhibit limited intra-class coverage. Also while the authors report the FID metric for different eval results, there is a risk of mode-collapse in the SG-Exemplar method where the generated samples are too close to the examples and not diverse enough. It will be good to provide a metric (for e.g. average pairwise distance of LPIPS in the feature space) on what is the intra-class diversity of the generated samples.
2) Latency and memory increase in SG-Text: Since there are M gradient updates per de-noising step (between t_start and t_end), and the authors show that M = 10 results in optimal performance in the appendix, it will be good to include what is the increase in latency and memory requirements during inference compared to other baselines, as this may be an important consideration factor for users deploying this method. Reporting time taken per image and peak GPU VRAM usage can be good metrics to track. Given that the gradients are calculated both for the UNet and the CLIP models, there might be a significant increase in memory usage.

---

> ### Author Response · Authors · 2026-01-25
>
> Thanks a lot for your thoughtful comments. Your insights have helped us in improving the analysis of the proposed method. Please find below the response to each comment.
>
> #### 1. [Robustness and Intra-Class Diversity]
> Thank you for these insightful suggestions. We address both concerns below.
>
> `[Robustness]`: As noted below Eq. 12, our anchor points (derived from exemplars) serve as estimates for conditional expectations. By the Law of Large Numbers, these estimates converge to the true conditional expectation as the number of exemplar samples increases. While theoretical convergence requires infinite samples, we empirically demonstrate that our method achieves robust performance even with modest sample sizes, showing consistent improvement with random sampling.
>
> We conducted systematic ablation studies on gender debiasing using the P2 model with randomly selected exemplar samples (averaged over three independent runs):
>
>
> | Number of Exemplar Samples ($k$) | FD      | FID   |
> |----------------------------|---------|-------|
> | 1                          | 0.241   | 80.66 |
> | 8                          | 0.00146 | 34.61 |
> | 20                         | 0.00092 | 32.53 |
> | 50                         | 0.00073 | 30.19 |
>
> These results validate our framework: both fairness (FD) and image quality (FID) improve consistently as $k$ increases. Notably, performance stabilizes beyond $k=8$, suggesting diminishing returns and justifying our default choice. The consistency across independent runs demonstrates robustness to sampling variance.
>
>
> `[Intra-Class Diversity]`Further, we compute LPIPS distance [1] as suggested to analyze the diversity of generated samples. Particularly, following [3,4], we assign 1000 generated images to one of the $k$ possible clusters (for $k$ exemplar samples) based on the lowest LPIPS distance [1]. Next, we compute the average pair-wise LPIPS metric among the members of the same cluster. Finally, we take the average over the k clusters. A method will have a zero score if it generates same image everytime. We provide the results with $k=8$ below:
>
> |Method       |Intra-cluster pairwise LPIPS distance|
> |-------------|--------|
> |Vanilla      |0.61    |
> |Balancing Act [2]|0.57    |
> |SG Exemplar  |0.64    |
> |SG Text      |0.66    |
>
> It can be seen that the LPIPS distance of SG-Exemplar closely outperforms the vanilla generation. We attribute this to two things: (i) explicit score guidance enabling exploration of additional modes beyond the original model distribution, and (ii) generation steps outside the guidance window, $\mathcal{T}$ encouraging diversity. Further, SG-Text achieves the highest diversity (0.66) as it avoids dependence on specific exemplars or classifiers.
>
> Further, to analyze the effect of $k$, we compute LPIPS metric for different values of $k$ in the following table:
>
> | Number of Exemplar Samples ($k$) | Intra-cluster pairwise LPIPS distance |
> |----------------------------|---------|
> | 1| 0.15 |
> | 8 | 0.64 |
> | 20 | 0.71 |
> | 50      | 0.74 |
>
> On expected lines, with just a single exemplar sample ($k=1$), there is indeed a mode-collapse since the generation is biased towards a single sample. However, with increasing $k$, LPIPS metric improves consistently because of the better estimate of conditional expectation as mentioned above.
>
>
> #### 2. [Latency and Memory Increase]
>
> Thank you for highlighting this point. We provide comprehensive measurements below (per-image, batch size 16, on NVIDIA A6000):
>
> |Method       |Time (s)| Peak GPU Memory Usage (GB) |
> |-------------|--------|----------------------------|
> |Vanilla      |1.736   |3.21                        |
> |Balancing Act [2]|9.427   |7.44           |
> |SG Exemplar  |4.211   |4.10                        |
> |SG Text      |12.941  |10.27                       |
>
> SG-Text incurs a $7.5\times$ latency increase and $3.2\times$ memory increase compared to vanilla generation, primarily due to $M$ gradient steps and backpropagation through both UNet and CLIP encoders for computing gradients.  For clarity, we will explicitly mention the increased latency and memory usage of SG-Text in our limitations.
>
> We also note that, computational cost is comparable to Balancing Act [2], while achieving superior performance across other metrics.
> Further, SG-Exemplar offers a more efficient alternative ($2.4\times$ latency, $1.3\times$ memory) while maintaining strong debiasing performance, making it suitable for resource-constrained deployments.
>
>
> We will incorporate these new analyses and results into the final version of the manuscript.
>
> [1] Zhang et al. "The unreasonable effectiveness of deep features as a perceptual metric." CVPR 2018. \
> [2] Parihar et al. "Balancing act: distribution-guided debiasing in diffusion models." CVPR 2024. \
> [3] Ojha et al. "Few-shot image generation via cross-domain correspondence." CVPR 2021. \
> [4] Mondal et al. "Few-shot cross-domain image generation via inference-time latent-code learning." ICLR 2023.

---

### Review · Reviewer_dkjP · 2026-03-06

**Summary Of Contributions:**

This paper proposes a training-free inference-time method called Score Guidance for debiasing diffusion models. The framework supports both text guidance and exemplar guidance, making it flexible across scenarios. Experiments show that this method has certain effectiveness. The authors are recommended to address the following comments to improve the significance of the study and for the benefit of a wider audience.

1. The derivation of Eq. (8) relies on the assumption that the attribute-conditional distribution can be expressed as a linear combination of mixture components. However, in real-world visual generation tasks, different attributes are often highly entangled, and this assumption may not strictly hold.

2. The paper focuses on a lightweight debiasing strategy. However, I noticed that a recent SOTA work [1] is not discussed. Including a discussion of this work in the Related Work section would help better contextualize the contribution.

3. From a technical perspective, I do not observe clear methodological weaknesses beyond the clarification questions mentioned above. However, a broader challenge common to fairness research remains: defining fairness for generative models is inherently context-dependent and often influenced by external factors beyond the model itself. The paper adopts a reasonable formulation based on balanced representation and comparable quality across groups, but whether this fully captures fairness in real-world scenarios remains an open question. I view this more as a discussion point rather than a concrete weakness.

[1] LightFair: Towards an Efficient Alternative for Fair T2I Diffusion via Debiasing Pre-trained Text Encoders. NeurIPS 2025.

**Audience:**

Yes

**Audience Explanation:**

The findings of this paper would be of clear interest to a substantial portion of TMLR's audience, particularly researchers working on generative modeling, responsible AI, and multimodal generation.

**Broader Impact Concerns:**

The paper aims to mitigate bias in diffusion models and therefore has a generally positive societal intent, particularly in improving fairness in generative systems. By enabling more balanced representation across demographic attributes, the proposed method could help reduce harmful stereotypes or imbalanced portrayals in generated content.

**Claims And Evidence:**

Yes

**Claims Explanation:**

The main claims of the paper are supported by clear, accurate, and convincing evidence. The authors provide both theoretical motivation and empirical evaluations across multiple settings, including binary, multi-class, and multi-attribute scenarios, as well as conditional generation using Stable Diffusion. The experiments compare the proposed method against several relevant baselines and use standard metrics such as FD and FID. The theoretical analysis and empirical results jointly provide strong and coherent support for the paper’s claims.

**Requested Changes:**

Please address the issues raised in the Summary of Contributions section.

---

> ### Author Response · Authors · 2026-03-07
>
> We sincerely thank the reviewer for their assessment of our work, acknowledging the flexibility of proposed Score Guidance framework across different modalities and its effectiveness. We deeply appreciate the constructive feedback, which helps strengthen the theoretical and contextual framing of our paper. Below, we address each of your points in detail:
>
> #### 1. [On Equation (8) and Attribute Entanglement]
> We thank the reviewer for this comment. We agree that in real-world, attributes are highly entangled (e.g., specific genders being statistically correlated with certain professions or races in the training data).
>
> However, we respectfully clarify that Eq. (8)  does not assume attributes are disentangled; rather, it represents exact marginal projection over a specific target attribute. When we formulate the unconditional score as a mixture over a single attribute set $A = \{a_1, a_2\}$ (e.g., gender), any other highly entangled attributes $B$ (e.g., profession) are implicitly marginalized out within the conditional score term $\nabla \log p(x_t | a_i)$. Mathematically, the conditional score naturally absorbs this entanglement:
>
> $$\nabla \log p(x_t | a_i) = \sum_j p_{train}(b_j | x_t, a_i) \nabla \log p(x_t | a_i, b_j)$$
>
> Therefore, the linear combination strictly holds for the marginal space of attribute $A$.
>
> We do, however, agree with the reviewer's underlying intuition: because of this entanglement, guiding solely on attribute $A$ will inadvertently leave the generated samples biased with respect to the entangled attribute $B$, as the SDE is pulled by the (possibly skewed) training prior $p_{train}(b_j | a_i)$.
>
> This is exactly why we designed our framework to handle multiple attributes simultaneously. As detailed in Section 4.1.2 and Appendix B.1, our method breaks this entanglement through a two-step mathematical guarantee:
>
> 1. `Recursive Tagging`: Instead of relying on the model's biased joint distribution, we construct a statistically independent joint reference $p_{ref}(a_i, b_j) = p_{ref}(a_i)p_{ref}(b_j)$. We recursively tag the batch to exactly match these joint proportions, bypassing the model's  $p_{train}(b_j|a_i)$ entirely.
> 2. `Alternating Projections`: We perform sequential score updates for $A$ and then $B$. Mathematically, this mirrors Projected Gradient Descent  on a composite potential $U_{joint}(x) = U_A(x) + U_B(x)$. By the theory of alternating projections, the predicted clean latent state $\hat{x}\_{0|t}$  is guaranteed to converge to the intersection of the attribute manifolds ($\mathcal{B}\_{a_i} \cap \mathcal{B}\_{b_j}$).
>
> Because the samples converge to these joint states, and the exact count of these states is deterministic, our method corrects the marginals of both $A$ and $B$, effectively disentangling them during inference. We will add a discussion in Section 3 to explicitly clarify this marginalization property and how the proposed multi-attribute framework resolves this issue.
>
> #### 2. [Contextualization with LightFair]
> We thank the reviewer for pointing out this excellent and recent work. We will gladly include LightFair in our revised Section 2.2 (Debiasing Diffusion Models) to better contextualize our contributions.
>
> LightFair provides an efficient debiasing method by directly mitigating biases within the pre-trained text encoders (e.g., CLIP) of Text-to-Image models. Consequently, its application is fundamentally restricted to conditional diffusion models. In contrast, our proposed Score Guidance (SG) is a generic framework successfully debiases both unconditional and conditional models without retraining, offering a more generalized intervention strategy.
>
> #### 3. [Definition of Fairness in Generative Models]
> We deeply appreciate the reviewer's nuanced perspective, and we strongly agree. While our paper adopts a mathematical formulation of fairness based on statistical demographic parity (via user-provided reference distributions $p_{ref}^a$), this algorithmic balancing is only one definition of bias. Real-world fairness is fundamentally socio-technical, highly context-dependent, and influenced by historical and cultural nuances that cannot be entirely solved by balancing generation ratios alone.
>
> Following your suggestion, we will add a "Broader Impacts and Limitations" paragraph to our Conclusion. We will explicitly acknowledge that while Score Guidance provides a robust, training-free debiasing method, deploying truly fair generative models requires embedding such tools within a broader, human-in-the-loop pipeline that actively considers the downstream societal context and potential harms of the generated media.

---

### Decision · Action_Editor_MUA8 · 2026-04-13

**Recommendation:** Accept as is

**Audience:**

Yes

**Audience Explanation:**

Fairness in diffusion models is an important, acute and active research problem. All reviewers agree.

**Claims And Evidence:**

Yes

**Claims Explanation:**

The paper proposes two variants of score guidance, a debiasing technique for diffusion model inference. The paper presents a comprehensive experiments across diffusion models, conditional classes, and conditioning modes (multi-class, multi-attribute).

All reviewers agree.

---

> ### Author Response · Authors · 2026-05-01
>
> Dear Editor,
>
> We would like to thank you for your time and effort in reviewing our manuscript. We have carefully considered the comments and suggestions provided by the reviewers, and we have made the necessary changes to improve the quality of camera-ready version of our paper. Below, we have addressed each comment in detail.
>
> ### Additional Results
> 1. **Debiasing results on SDv2.0 and SDXL** (Reviewer TCYy): New results on SDv2.0 and SDXL have been added in Appendix E, where we have added quantitative and qualitative results using SG - Text and SG - Exemplar.
> 2. **Ablation Result on Number of Exemplar Samples** (Reviewer Kg9R):   We have added new results analyzing the effect of the number of exemplar samples in Appendix C.1.2. Quanititative results illustrating the effect of the number of exemplar samples on debiasing performance, generationg quality and generation diversity have been added in Table 13, 14, and 15.
> 3. **Computational Requirements and Latency Analysis** (Reviewer Kg9R): We have added a new section in Appendix C.2 that provides analysis of the computational requirements and latency of our proposed method. We have included a comparison of the computational resources needed for our method compared to baseline methods, as well as an analysis of the latency introduced by our method during inference.
>
> ### Additional Clarifications
> 1. **Defitinition of** $\nabla_{h^{(i)}}$ (Reviewer TCYy): We have added a clarification regarding the definition of $\nabla_{h^{(i)}}$ in the main text below Equation 10.
> 2. **Attribute Entanglement** (Reviewer dkjP, TCYy): We have added a new subsection under Section 3.2 that clarifies the effect of debiasing a single attribute on other attributes. We have included a discussion on the potential for attribute entanglement and how our method addresses this issue.
> 3. **Definition of Fairness** (Reviewer dkjP): We have revised the Broader Impact Statement to include nuances regarding the definition of fairness. We have acknowledged that fairness is a complex and multifaceted concept that can be defined in various ways depending on the context. We have emphasized the importance of considering different perspectives on fairness and the need for ongoing research in this area.
>
> We believe that these additions and clarifications have significantly improved the quality of our manuscript, and we are grateful for the time and constructive feedback provided by the reviewers.